# Lithography for robust and editable atomic-scale silicon devices and memories

Roshan Achal[1,2], Mohammad Rashidi [1,2], Jeremiah Croshaw[1], David Churchill[3], Marco Taucer[1,2], Taleana Huff[1,2], Martin Cloutier[4], Jason Pitters[2,4] & Robert A. Wolkow [1,2,4]

At the atomic scale, there has always been a trade-off between the ease of fabrication of structures and their thermal stability. Complex structures that are created effortlessly often disorder above cryogenic conditions. Conversely, systems with high thermal stability do not generally permit the same degree of complex manipulations. Here, we report scanning tunneling microscope (STM) techniques to substantially improve automated hydrogen lithography (HL) on silicon, and to transform state-of-the-art hydrogen repassivation into an efficient, accessible error correction/editing tool relative to existing chemical and mechanical methods. These techniques are readily adapted to many STMs, together enabling fabrication of error-free, room-temperature stable structures of unprecedented size. We created two rewriteable atomic memories (1.1 petabits per $in^2$), storing the alphabet letter-by-letter in 8 bits and a piece of music in 192 bits. With HL no longer faced with this trade-off, practical silicon-based atomic-scale devices are poised to make rapid advances towards their full potential.

[1] Department of Physics, University of Alberta, Edmonton, AB T6G 2E1, Canada. [2] Quantum Silicon, Inc., Edmonton, AB T6G 2M9, Canada. [3] Memorial University of Newfoundland, St. John's, NL A1B 3X5, Canada. [4] Nanotechnology Research Centre, National Research Council of Canada, Edmonton, AB T6G 2M9, Canada. Correspondence and requests for materials should be addressed to R.A. (email: achal@ualberta.ca)

The first demonstration of controlled atomic manipulation[1] fostered a vision of practical atomic-scale devices. Today, with recent innovations exhibiting control over approximately 8000 atomic sites[2] this vision is slowly materializing. Yet realization of functional atomic-scale devices outside of the laboratory has been limited by their instability at room-temperature[1,2] and their poor electronic isolation from supporting substrates[3,4]. Hydrogen lithography (HL), the removal of hydrogen atoms (depassivation), on hydrogen-passivated silicon surfaces is becoming an important technique in next-generation device designs[5–11]. These designs have potential to surmount both limitations[12–14] without introducing materials incompatible with current semiconductor fabrication processes[15–17]. Many disruptive applications have been proposed based on HL such as single atom transistors[6], quantum computing platforms[7,9–11], and atomic-scale logic devices[4,8], drawing both scientific and commercial interest alike. Several companies have even formed upon this and related techniques[11,14,17]. Overall device development has been delayed, however, by the inability of HL to fabricate large error-free atomic-scale structures[5,16–21], increasing the need for reliable error correction techniques.

The atomically precise removal of hydrogen from hydrogen-passivated silicon surfaces via a scanning tunneling microscope (STM) tip has been studied extensively for HL[16–18,22,23]. A select Si–H bond is broken using low energy inelastic electron scattering, exposing a dangling bond (DB) of the underlying silicon atom[16,22,23]. The DB can be thought of as an atomic quantum dot, capable of localizing two electrons, with its electronic states isolated within the silicon band gap[19,24]. DBs present an ideal platform for new technological applications[6–11,19,25] because of this electronic isolation and their stability at temperatures near 500 K[12,13], overcoming two major hurdles for practical implementation of atomic-scale devices[4]. Beyond that, the placement of DBs with atomic precision and their ability to couple to one another enables delicate control over their properties and electronic occupation[8,26], permitting engineering of quantum states[9,19] and artificial molecules[4,19]. Precise HL also offers a route to other atomic-scale and molecular devices through atomically site-selective chemical templating[4,6,7,10,11].

Although HL has been incrementally improving over time, the path towards perfect large-scale structures and the full realization of silicon DB-based devices has been unclear[15–17,19,20,23]. Increasing atomic precision during HL was the main focus of efforts[20], since any hydrogen removed in error could render structures inoperable. Non-site-specific chemical repassivation of DBs has been observed[27], but without the means to deterministically repair mistakes, this strict requirement of 0% error has hindered the development and prototyping of novel device designs[4,6,8,28]. A means to controllably correct errors was recently shown using a cryogenic atomic force microscope (AFM), where individual DBs were repassivated with a hydrogen-functionalized tip[5,14]. While a striking demonstration of atomic control, the utility of this technique is limited as the reported repassivation procedure is slow (approximately 10 s per DB)[14], reducing its practicality for larger structures. The frequency shift signal utilized in AFMs requires two independent feedback loops, restricting the maximum speed of the overall process. This procedure is further slowed by the need to re-functionalize the tip with hydrogen in between each event[5,14]. Existing methods of hydrogen deposition onto a silicon surface with an STM tip[29,30] are unsuitable to edit or correct DB-based devices. They involve harsh parameters capable of damaging fabricated structures or the tip itself[20].

Here, working at 4.5 K, we demonstrate straightforward STM methods for automated HL and hydrogen repassivation (HR), which do not significantly alter tip structure. Controlled application of voltage pulses is used for lithography, while repassivation uses linear tip movement towards the sample surface under small applied bias voltages. In both techniques, the STM feedback controls are suspended, employing changes in tunneling current as the only signal. We discovered two unique signatures in the STM tunneling current associated with successful HR events. When implemented as control signals, they will increase the viability of rapid fully automated error correction. In conjunction, HL and HR techniques substantially advance our fabrication abilities towards functional atomic-scale devices. To illustrate their precision and practicality, we created functional 8-bit and 192-bit rewritable atomic memory blocks from DBs with the highest reported storage density to date[2,28].

Interest in atomic memory has been reignited with foundational work on chlorine-passivated Cu(100), establishing a sophisticated scanned probe architecture to create a kilobyte of memory from surface vacancies, without the need to vertically manipulate atoms[2]. The memory operates near 77 K, where it remains stable for at least 44 h[2]. There are several key features of DB-based memories that allow us to push atomic-scale storage even further than this already substantial progress. Patterned DB structures exhibit improved thermal stability, remaining stable for an additional 400 K above liquid nitrogen temperature[12,13]. The maximum storage density of memory blocks can be increased by 32%, as DBs can be placed in close proximity to one another. In addition to density, the number of available bits is not predetermined at the time of sample preparation[2]. DBs can now be created or removed as needed using HL and HR (assuming a sufficient supply of hydrogen atoms), theoretically allowing the entire hydrogen-terminated surface to be written to. Creating additional vacancies/bits in the Cu(100) system is currently not possible without damaging the STM tip[2]. Furthermore, a semiconducting substrate as opposed to a metallic one opens the possibility of interfacing with integrated electronics, whether at the atomic-scale or using more conventional nanofabrication.

## Results

**Hydrogen lithography.** To begin automated HL, the location of every hydrogen atom in a select area needs to be determined for accurate STM tip registration during fabrication. Slight errors in the tip position can result in incorrect atoms being removed. Fast, fully autonomous lithography also requires the location of each atom to be known, such that the surface does not need to be reimaged after each removal event to determine the next site. The periodicity of the hydrogen-passivated Si(100)-2×1 surface (Fig. 1a) permits the location of every hydrogen atom to be determined from a single STM image (while accounting for nonlinearities in the scanner) through the use of Fourier analysis[31] (Fig. 1b–f). Such an analysis is relatively immune to the presence of small surface defects and DBs due to their spatially localized nature in images compared to the extended periodicity of the surface itself. Figure 1 illustrates the basic features of this process. In practice, we use images between $10 \times 10$ and $40 \times 40$ nm$^2$ to determine the location of the hydrogen atoms on a given sample terrace. Once the surface has been characterized, the desired pattern is mapped onto the lattice. Next, the tip is brought over each lattice site of the pattern and 20 ms voltage pulses between 1.8 and 3.0 V are applied at a fixed tip-height ($V = 1.4$ V, $I = 50$ pA) until the successful removal of hydrogen has been detected. Figure 2a, b, d shows the process of building DB structures (Fig. 2e) with HL while using HR to correct errors (Fig. 2c). Unlike conventional HL[16,32], the STM feedback control is disabled during the voltage pulses. This allows jumps in the tunneling current to be used as an indicator of success[15], which

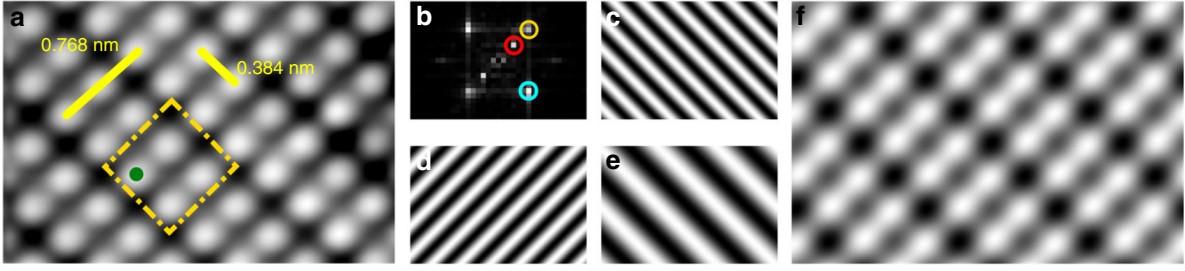

**Fig. 1** Surface geometry of hydrogen-passivated Si(100)-2×1. **a** ($V = 1.4$ V, $I = 50$ pA, $T = 4.5$ K, $2 \times 3$ nm$^2$) Scanning tunneling microscope (STM) image of hydrogen atoms bonded to the Si(100)-2×1 surface. The distance between identical sites along and across a dimer row are shown in yellow (solid). The surface geometry allows for the creation of single atom bits at ultra-high densities. The area of one bit (including spaces between atoms), 0.590 nm$^2$, is outlined in yellow (dashed), and is defined as a binary zero. The hydrogen atom denoted with a dot (green) is removed with an STM tip to create a dangling bond, changing the bit to a binary one. **b** 2D-Fourier transform (power spectrum) of **a**, where the three dominant spatial frequencies have been isolated. **c**–**e** Filtered STM images of **a**, for each dominant frequency (peak) shown in **b**, **c** bottom peak (blue), **d** top peak (yellow), **e** middle peak (red). **f** The resulting image from the sum of frequencies shown in **c**–**e**, reconstructing the essential features of the STM image shown in **a**, allowing for the position of each surface hydrogen atom to be determined. Scale bar, 1 nm

can be detected faster and more accurately than jumps in tip-height through the feedback circuitry (see Methods).

Using this procedure, the probability of detrimental uncontrolled apex changes is low. By beginning removal attempts at 1.8 V (see Methods), higher voltages, which are more likely to change or damage the tip, are only reached when necessary. Conservatively, on the order of 10 DBs can be created consecutively without some type of minor modification to the tip. However, we have found that HL efficiency is not particularly sensitive to minor changes of the tip, so the actual number of DBs that can be created without altering removal efficiency during fabrication is often larger. Should the tip change so much that it is no longer suitable for HL purposes, an automated tip forming routine can be called to recondition the tip through controlled contact with the surface[33]. This routine takes advantage of a machine learning algorithm, and STM image data for training sets, to automatically identify the quality of the probe by imaging a DB, initiating reconditioning when necessary[33].

An important consideration inherent in all scanned probe lithography is the existence of thermal drift and creep, both of which can also cause uncertainty in the position of the tip, leading to errors. At 4.5 K, these factors can be well controlled by allowing the STM to stabilize over a period of several hours. However, at warmer temperatures or in situations where allowing the STM to stabilize is not an option, a more active solution is required. To address these factors, we implemented periodic image realignments into the HL workflow. Before initiating the HL procedure, an area near the lithography location (around $10 \times 10$ nm$^2$) is imaged as a reference. After a set time, lithography is paused, and this area is reimaged to determine how much the tip has been offset from its intended position due to creep and drift. The remaining sites in the pattern are shifted appropriately to compensate and lithography resumes. The effectiveness of this realignment can be increased by reducing the interval between reference checks, permitting an optimization between speed and accuracy depending on a given application. We found that without realignment the lithographic accuracy during HL using a non-stabilized STM was near 35% for a particular structure. Under the same conditions using moderate active realignment it was over 85%, which is within a suitable range to then correct the remaining errors using HR.

**Hydrogen repassivation.** To correct errors, we use the STM tip to address individual lattice sites with atomic precision to repassivate select DBs (Supplementary Figure 1). Ab initio calculations

have revealed that certain silicon terminated tips with a hydrogen atom bonded to the apex (functionalized) are capable of repassivating DBs[34]. Through controlled contact of the STM tip with the surface during tip-conditioning, silicon atoms may become affixed to the tip to form the necessary structure for HR[5,14,34]. When the tip becomes functionalized with a hydrogen atom, there is a change in STM imaging resolution[5,14]. With a functionalized tip, the first step of HR is to position it over a DB at a sample voltage of 1.4 V and current of 50 pA. The feedback control is then disabled, and the sample voltage is changed to a value between 100 mV and 1.0 V. While recording the tunneling current, the STM tip is brought 500–800 pm towards the sample, then is retracted to its original position. The voltage is reset to the original value of 1.4 V and the feedback control is restored. This entire process, once a user has selected a site, has been automated, taking approximately 1 s. It can be initiated at the press of a button and repeated until a successful repassivation signature is observed. Work is underway to integrate this new HR process within the HL workflow to enable fully autonomous fabrication and correction. Errors will be automatically detected via image recognition, and subsequently corrected using the HR technique (see Methods).

A necessary element for the implementation of fully automated HR is the existence of a reproducible, easily distinguished, signature indicative of successful repassivation, much like the jump in tunneling current used during HL. We usually observe a sudden increase in the tunneling current while approaching the surface when a DB is successfully repassivated that is not present as the tip retracts (Fig. 3a). The measured current is related to the overlap of the imaging orbital of the tip, and orbitals of features on the surface[16,35]. We associate this signature (type-I) with the removal of a hydrogen at the tip apex and a return to pre-hydrogen-functionalized imaging resolution. The increase in current is possibly due to the new apex orbital having a larger spatial extent, creating a greater overlap between the tip and sample surface compared to that between a DB and a hydrogen-functionalized tip (held at the same tip–sample separation).

Following the same HR procedure with a tip that is not functionalized, we have found it is also possible to repassivate DBs with no associated change in imaging resolution. Since STM imaging resolution is particularly sensitive to changes in the tip apex orbital[14,16,35], we assume the imaging orbital is unchanged and that the hydrogen atom comes from the off-apex region. In addition to functionalization, there are several mechanisms through which atomic hydrogen may be available on the (off-apex) surface of a tip for HR. The surface of silicon terminated

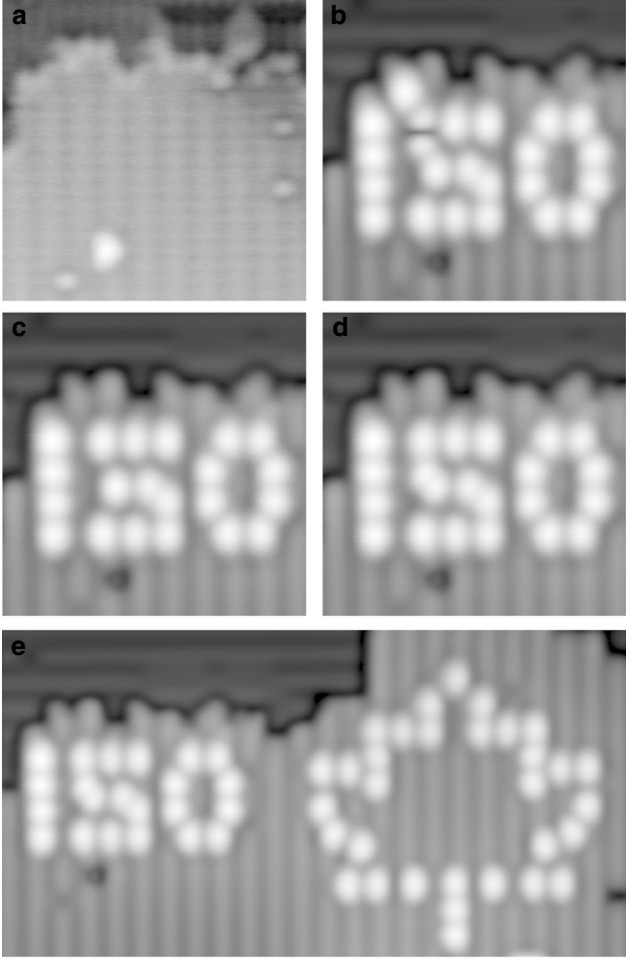

**Fig. 2** Fabrication and correction on hydrogen-terminated Si(100)-2×1 via hydrogen lithography and hydrogen repassivation. **a** ($V = 1.4$ V, $I = 50$ pA, $T = 4.5$ K, $10 \times 10$ nm$^2$) Scanning tunneling microscope image of the fabrication area used to determine the location of each hydrogen atom through Fourier image analysis. **b** ($V = -1.7$ V, $I = 50$ pA, $T = 4.5$ K, $10 \times 10$ nm$^2$) Fabrication of the characters "150" using hydrogen lithography (HL), one atom has been removed in error above the characters and one atom is incorrectly placed inside the "5". The created dangling bonds (DBs) appear as bright protrusions. **c**, **d** Using the hydrogen repassivation technique, the errors were subsequently erased, then the correct hydrogen atoms were removed with HL to create an error-free structure. **e** ($V = -1.7$ V, $I = 50$ pA, $T = 4.5$ K, $11 \times 21$ nm$^2$) The completed structure of 54 DBs, depicting the characters "150" and a maple leaf. Scale bar, 5 nm

tips can become passivated with hydrogen[34], allowing atomic hydrogen to physisorb on top[14] (see Supplementary Note 1, Supplementary Figure 2). If the tip is not silicon terminated, it has been shown that hydrogen is also capable of physisorbing on the surface of some metallic tips[29,30,36]. Since the precise structure of the tip is unknown, the available hydrogen on the tip is likely due to a combination of these mechanisms. We have found that the STM tip can be loaded with hydrogen atoms through the creation of several DBs (far from the current structure) using HL[14]. We observed that an average of three DBs (to a maximum of five) can be repassivated successively, without the need to load the tip with additional hydrogen (Fig. 4a). During HR, when the DB is repassivated with an off-apex hydrogen we see a second signature in the tunneling current (type-II), a sudden decrease (Fig. 3b) (also observed during HR at 77 K, Supplementary Figure 3). The sudden drop in current is due to a reduction in overlap between

the tip and sample. DBs appear as bright protrusions on the sample surface (Supplementary Figure 1b) compared to the surrounding hydrogen (orbital more spatially extended towards tip). There is a decrease in the size of the surface orbital after the DB has been repassivated, which reduces overlap/current, as the tip orbital remains unaltered.

Unlike the type-I process, which theoretically relies on a particular tip state to enable the transfer of hydrogen[34], the type-II process appears to be much less restrictive. We have been able to observe type-II HR events even after impressing the tip approximately 1 nm into the sample surface. Both processes have been observed and reproduced on a number of physically different tungsten tips, during the fabrication of numerous different structures. The structures in Figs. 2 and 4 were created using different tips for example. We recorded the location of type-I and type-II signatures in the tunneling current for 119 successful HR events collected using seven different tips (see Supplementary Figure 4 for additional recordings). Figure 3c shows the distribution of distances the tip traveled towards the surface for HR to occur. The majority of events (around 90%) occur before moving 550 pm. Closer tip approaches have an increased tendency to change the tip structure. This value provides an ideal parameter for fully automated HR, optimizing the probability of repassivation while mitigating that of harmful apex changes.

During our observations, we noted that when a tip is hydrogen-functionalized, as indicated by a change in STM imaging resolution, it is still possible to transfer an off-apex hydrogen to the surface (type-II signature) without altering the apex itself (leaving the tip functionalized). That is, with a hydrogen-functionalized tip, it is not guaranteed to first remove the apex atom and observe a type-I HR signature, as an off-apex hydrogen may be more mobile and easily transferred to the surface first, causing a type-II HR signature. We have also never observed sequential type-I signatures without deliberately re-functionalizing the tip in between HR attempts, suggesting the tip is unable to functionalize spontaneously with off-apex hydrogen. This observation is consistent with experimental imaging data, where spontaneous changes in image resolution with a tip suitable for HL/HR are rare.

With the ability to know when a tip is hydrogen-functionalized, and when the apex atom has been removed, it may now be possible to correlate d$I$/d$V$ spectroscopy curves over hydrogen-terminated silicon with the specific tip states necessary for hydrogen transfer to occur[34] (type-I). If such a correlation is found, d$I$/d$V$ spectroscopy would provide a new, more rapidly acquirable metric to determine if the tip is suitable for HR. These curves could then be used as training data in the automated tip forming routine to always ensure the tip is in the required state, without the need to take an entire STM image[33].

**Atomic-scale memories**. We used HL combined with HR to create two different working atomic-scale memories (Fig. 4). We defined a bit with four lattice sites, giving a one atom buffer between neighboring DBs (Fig. 1a). Due to the ideal geometry of the hydrogen-passivated Si(100)-2×1 surface, this arrangement yields an extremely high bit density of 1.70 bits per nm$^2$. Conservatively, estimating a 20% reduction in storage density to integrate practical control architecture, this system still provides the highest density solid-state storage medium available. We used the 8-bit memory to sequentially encode the ASCII binary representation of each letter of the English alphabet, overwriting the previous letter each time (Fig. 4a). Writing each letter took between 10 and 120 s, depending on how many DBs needed to be created and repassivated. Random changes in the tip apex

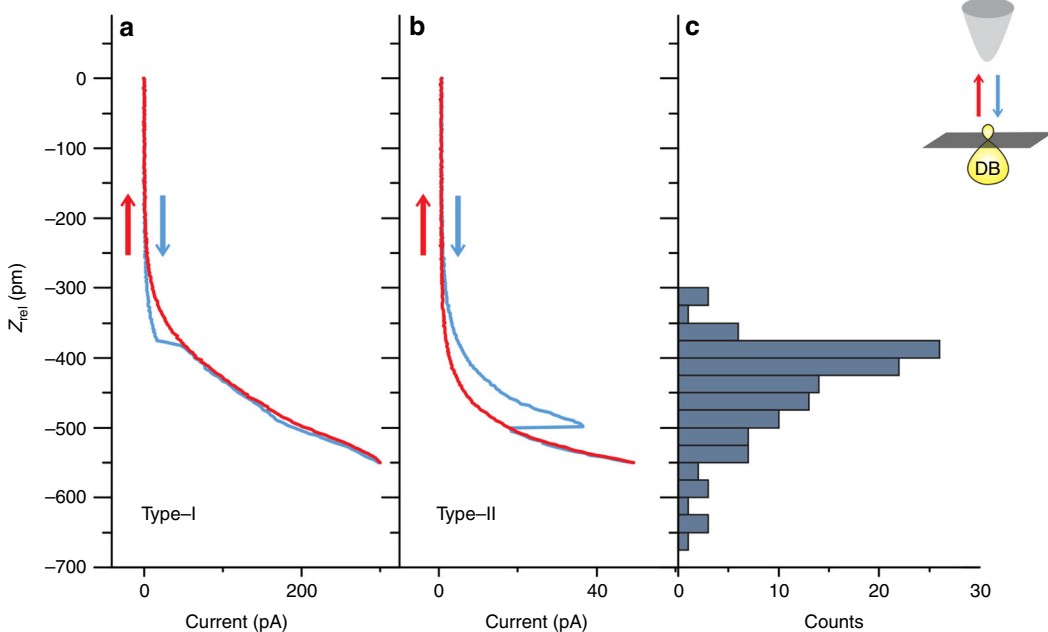

**Fig. 3** Tunneling current signatures of hydrogen repassivation and the number of successful events with tip approach distance. **a** ($V = 0.4$ V, $T = 4.5$ K) The recorded tunneling current as the scanning tunneling microscope (STM) tip (set over a dangling bond (DB) at 1.4 V and 50 pA) is brought towards the surface (blue) and as the STM tip is retracted (red) during hydrogen repassivation (HR). A sudden increase as the tip approaches the surface can be observed in the tunneling current (blue), indicating a successful HR event. This signature (type-I) is typically associated with a change in STM imaging resolution, and a removal of hydrogen from the tip apex. **b** ($V = 0.2$ V) A second distinct signature (type-II) has also been observed during HR, with a sudden decrease in tunneling current during the approach towards the surface (blue). This type-II signature is not typically associated with a change in STM imaging resolution. **c** The tip approach distance where either signature occurred was recorded for 119 unique HR events. Nearly 90% of events occur before 550 pm. The inset depicts the STM tip approaching a DB on the surface

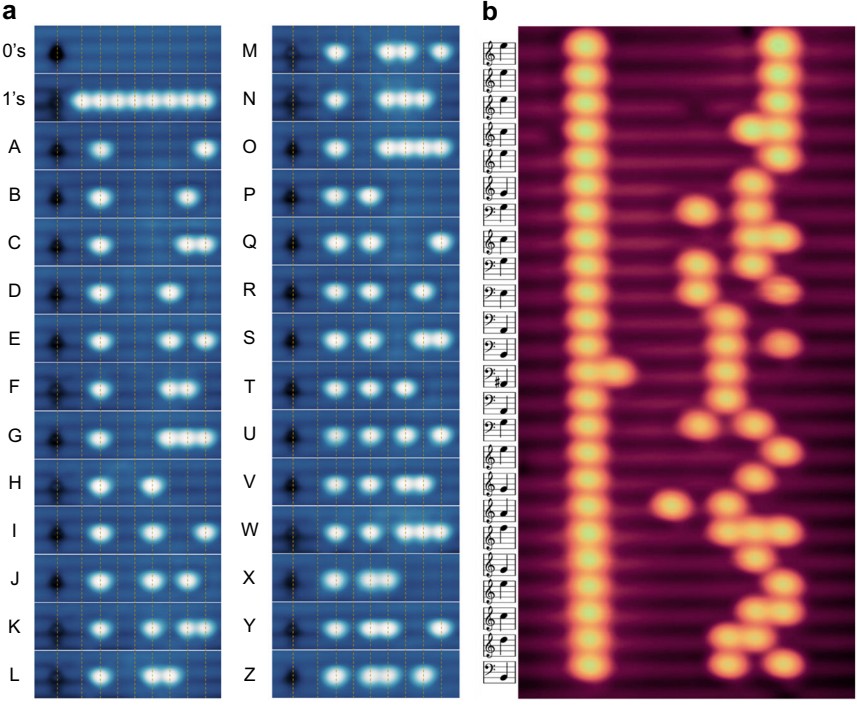

**Fig. 4** 8-bit and 192-bit atomic rewritable memories. **a** ($V = -1.75$ V, $I = 50$ pA, $T = 4.5$ K, $2.4 \times 10$ nm$^2$) Scanning tunneling microscope (STM) images of a rewritable 8-bit memory constructed from dangling bonds (DBs). The DBs appear as bright protrusions in the images, the black defect is an area of missing silicon. The letters of the English alphabet were converted to their ASCII binary forms (e.g., A = 01000001) and written to the same area using hydrogen lithography and hydrogen repassivation techniques. An average of three bits could be repassivated sequentially between the letters shown. Scale bar, 3 nm. **b** ($V = -1.8$ V, $I = 50$ pA, $T = 4.5$ K, $21.5 \times 10.7$ nm$^2$) An STM image of an expanded 192-bit memory, storing 24 simplified notes (converted into binary) of the popular Mario videogame theme song. The notes consist of 62 DBs, forming the largest error-free DB structure to date. Both memories have a bit density of 1.70 bits per nm$^2$ and can be read directly from the STM or from a stored image. Scale bar, 4 nm

occurred infrequently, altering HL and HR efficiencies (remedied through controlled contact of the STM tip with the surface to condition the apex structure). The HR stage is currently the slowest step, limited by the number of available hydrogen atoms on the surface of the tip. Moving away from the structure to reload the tip after repassivating several DBs introduced a significant delay. This may only be a factor for structures with a small number of DBs, like those we have presented. With structures requiring more DBs, there will be a continued source of hydrogen to the tip as each new DB is created (equivalent to the reloading procedure). With enhanced automation to incorporate periodic intervals for HR/error correction during HL, the need to travel away from the structure to reload the tip can be reduced or altogether eliminated. Further improvements to HR speeds may be possible through tip materials like platinum, which is able to hold at least 1000 atoms of hydrogen on its surface[29].

We created a larger 192-bit memory with the same bit density, encoding the iconic notes (simplified) of the Mario videogame theme song (Fig. 4b). This 62 DB arrangement took 250 s to create, and represents a leap forward in HL capability. We can read the 192-bit memory directly from the STM as the tip scans over the structure, or later from recorded images (see Supplementary Movie 1). The room-temperature stability of DB structures has already been well established in prior works[4,17,18,26,28]. DBs on the surface of silicon are known to face barriers to diffusion in excess of 1.4 eV in either direction[12,13]. Taking an exponential prefactor of $A = 2 \cdot 10^{14} \, s^{-1}$ and the lowest energy barrier of 1.46 eV[13] (intradimer diffusion), a DB is expected to experience only one hop in over 500 years due to thermally activated processes at 300 K. Such stability and density make DB-based memories a unique candidate for data archival and long-term storage.

Held at 4.5 K, we detected no unintentional changes in the memory at the end of 72 h of observation (Supplementary Figure 5). In this environment, we have worked with samples that have shown no significant surface degradation after half a year. The ultra-high vacuum (UHV) requirements may also eventually be relaxed as isolated DBs can be protected against spontaneous reactions through the appropriate choice of doping level of the silicon substrate. Highly doped n-type silicon results in negatively charged DBs, which have a barrier to reaction with closed shell species[37]. There is also only a subset of entities that are known to readily react with DBs[27,37–39]. Molecular hydrogen, which is commonly present in vacuum environments, requires two directly adjacent DBs along a dimer row for adsorption[40], reducing the likelihood of spontaneous repassivation of isolated DBs at greater separations (as in the memory) due to ambient gas. The inability to eliminate all naturally occurring DBs during sample preparation in an environment of $1 \cdot 10^{-6}$ Torr of hydrogen gas, or with intentional chemical dosing further supports this notion[27].

## Discussion

While these new HL and HR techniques are in their infancy, we demonstrated their immediate utility and applications by creating and editing large error-free DB structures through accessible STM-based means. Both these techniques can be implemented on many STMs with no modifications, operating over a range of temperatures. Though the exact parameters reported here are specific to hydrogen-terminated silicon, they can be adapted to other chemically similar systems such as hydrogen-terminated germanium[41,42], hydrogen-terminated diamond[41,43], and chlorine-terminated silicon[44]. There is also the potential to eventually extend these protocols to room temperature, as forms of HL have already been successfully demonstrated

there[16,20,23,26], and the uncontrolled transfer of hydrogen from an STM tip to a DB has been observed for HR[39]. However, fabricating structures/devices at cryogenic temperatures (where creep and thermal drift are not as pronounced) for use at room-temperature may already be sufficient for many applications. Further, the high-temperature stability of DBs removes one of the logistical issues surrounding the transportation of fabricated nano-devices to an end user, regardless of the temperature required for their operation.

Together, HL and HR unlock an array of new possibilities including the creation of hundreds of precisely placed identical qubits for quantum computation[9], the realization of room-temperature stable atomic-scale memory, and the long sought ability to controllably create regions with no DBs[27]. HR can also augment the yield of existing commercial fabrication methods for multi-atom-wide DB structures, where patterned edges are prone to errors[17,20,21]. As these HL/HR techniques mature, they will play an integral role in accelerating the progress of many of the proposed disruptive technologies[5–8,10,11]. Supported with methods of preserving the surface outside of vacuum[15,45], along with efforts in parallelizing scanned probe fabrication[46], these techniques bring the vision of practical atomic-scale devices one step closer to reality.

## Methods

**Experimental setup**. We took all measurements with a commercial low-temperature Omicron LT-STM operating at either 4.5 or 77 K. The base pressure inside the STM ranged from $3 \cdot 10^{-11}$ to $7 \cdot 10^{-11}$ Torr. The STM tips consisted of a polycrystalline tungsten wire (50 μm diameter), electrochemically etched in a solution of 2 M NaOH. The tips were then processed under UHV conditions to further clean and sharpen them[47].

**Sample preparation**. We first degassed a highly arsenic-doped Si(100) (0.003–0.004 Ω-cm) sample at 600 °C in UHV for 24 h to remove any adsorbed water. We then resistively heated the sample via rapid flashes to 1250 °C several times to remove all native oxide. Following that, we exposed the sample to $1 \cdot 10^{-6}$ Torr of hydrogen gas. A nearby tungsten filament held at 1900 °C was used to crack the gas into atomic hydrogen. We exposed the sample to the gas for 120 s with no heating, then rapidly flashed it to 1250 °C, after which we quickly brought the temperature down to 330 °C for 150 s to achieve the desired hydrogen-terminated $2 \times 1$ surface reconstruction. The sample remains in the preparation chamber for up to 15 min as the pressure slowly returns towards the initial base pressure.

**Automated hydrogen lithography**. The HL program was designed in-house, including a graphical user interface for atomic pattern input. An artifact-free positive sample voltage STM image of the working area is first analyzed to determine the position of each atom through a 2D-Fourier transformation (2DFT), extracting the dominant spatial frequencies of the surface from the power spectrum. This method accounts for nonlinearities in the STM motors as each directional motor ($x$ and $y$) may have slightly different responses to an applied voltage, which can be recovered in the spatial frequencies. Additionally, the angle of each sample may differ slightly, and this information is also present within these 2DFT data. After the surface is characterized, a desired device design or pattern may be input via the graphical user interface to be mapped onto the surface. Once a pattern has been mapped onto each hydrogen atom the HL procedure initiates. With the STM feedback controls on ($I = 50$ pA, $V = 1.4$ V) the tip is brought over the first site in the pattern. The feedback controls are then suspended, fixing the tip-height over the site. The tunneling current is recorded for reference, and approximately 20 ms voltage pulses are applied in the range of 1.8–3.0 V. The number of attempts ($N$) at each voltage, and the voltage increment can be set beforehand. Typical values are $N = 10$ with a 0.01 V increment. During the voltage pulse phase, the tunneling current is sampled immediately after each applied pulse and compared to the reference value. If the measured current is 60% larger than the reference value, it is deemed a successful hydrogen removal event. With this technique, false detections have been brought well below 0.5%, with most fabrication runs producing no false detections. After a successful hydrogen removal, or after the maximum number of pulses is reached (unsuccessful removal), the feedback controls are restored, and the tip is brought to the next site in the pattern (following a raster path) where the process repeats.

In an effort to better control drift and creep, after a set number of HL events a routine can be called to check tip alignment with a reference STM image. If any misalignment is detected[48], the remaining pattern is shifted appropriately so the next locations for atom removal are again centered over their corresponding atom.

The patterning–realignment cycle can continue until the pattern is complete. The image realignment can detect sub-nanometer shifts between images.

**Automated hydrogen repassivation**. The tip is set directly over the lattice site where a DB is present ($I = 50$ pA, $V = 1.4$ V). The feedback controls are switched off, locking the tip height. The sample voltage is changed to a value between 100 mV and 1.0 V, and then the tip is moved linearly towards the sample surface while recording the tunneling current. After the tip has traveled a distance of 550 pm towards the surface it is retracted to its original position. The original parameters are re-established, and the feedback control is restored. To date, no significant correlation between voltage and HR efficiency has been observed. The choice of voltage serves to limit the tunneling current to within ranges that prevent significant tip apex changes, while still providing adequate feedback signals. Typically, we perform HR at a bias of 200 mV and only adjust this value in the program when the signal falls outside of the desired range (3–300 pA). The strength of the signatures depends on the exact structure of the apex, as they can vary by an order of magnitude at the same applied bias (Supplementary Figure 3, 4). Even though the strength of the signatures vary, their shape remains characteristic, making them ideal for the detection of successful events (Supplementary Figure 3, 4). If the initial HR attempt is unsuccessful, the process can be repeated until a type-I or type-II signature is detected. Work is in progress to include automatic error detection after HL using image recognition to define arbitrary groups of sites for HR in an image. This will eventually enable fully automated HR, without any user intervention to select individual sites to initiate the HR process.

**Memory readout/image recognition**. The memory readout was accomplished with the use of Python and Open CV. The periodicity of the hydrogen-terminated Si(100)-2×1 surface allows for a grid to be defined over the surface, where each bit is contained within one cell. The image contrast of when a DB is present or not lends to threshold detection to determine if the bit is one or zero. Memories can be read directly from recorded STM images, or as the image is built up while the STM scans over the surface, populating each cell element. Readout speed is limited by the maximum STM scan speed. The musical playback was created with the use of the Pygame package, where detected bit patterns can be mapped into notes for playback.

**Data availability**. These data presented in this manuscript are available from the corresponding author upon reasonable request.

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

## Acknowledgments

The authors thank M. Salomons for his technical assistance, K. Gordon for helpful discussions, and J. Phillips for editing and proofreading the manuscript. We would also like to thank NSERC, AITF, NRC, and QSi for their financial support.

## Author contributions

R.A. conceived the memory design and hydrogen repassivation procedure. R.A., M.R., and J.C. performed the experiments to test the procedure and collect hydrogen repassivation curves. R.A., T.H., and M.R. discussed methods of repassivation. R.A. fabricated all structures presented in the manuscript. M.R., R.A., and J.C. developed automated hydrogen repassivation. J.P., M.C., M.T., M.R., R.A., and T.H. developed automated hydrogen lithography software. D.C. and R.A. developed memory readout. R.W. supervised the project. R.A. and M.R. prepared the manuscript. All authors participated in the review of the manuscript.

## Additional information

**Competing interests:** Patent applications are in process related to the subject presented in the manuscript. Some of the authors (R.A., M.R., M.T., T.H., J.P., R.A.W.) are affiliated with Quantum Silicon Inc. (QSi). QSi is seeking to commercialize atomic silicon quantum dot-based technologies. The remaining authors declare no competing interests.

