## [Peer Review File · Nature Communications]

Reviewers' comments:

Reviewer #1 (Remarks to the Author):

The manuscript "Lithography for Robust, Editable Atomic-scale Silicon Devices and Memories" by Roshan Achal et al. reports on improvements in STM-based lithography technique used for atomically precise formation of dangling-bond (DB) structures on a hydrogen passivated Si(100) surface under UHV conditions and at cryogenic temperatures. Authors introduce a protocol for desorption of single hydrogen atoms from the surface, which can be realized in an automatic manner. Moreover, Authors show novel STM-based method of repassivation of Si dangling bonds by adsorption of single H atoms, which can serve as error-correction or editing tool for the hydrogen lithography. By combination of these two approaches it is demonstrated that several atomically defined patterns of DBs can be formed, with the largest one possessing 62 DBs. Finally, Authors show rewritable pattern of 8 DBs, which serves as 8-bit memory able to store the alphabet letters.

In my opinion the presented results are an important technical step towards upscaling of atomically precise formation of DB patterns on hydrogen passivated semiconductor surfaces. Thus, the work should be of high importance to the community working in this particular field. However, I have doubts if the manuscript will have immediate interest to a broader audience as described protocols are very specific for Si(100):H surface. Additionally I have concerns about scalability of the presented lithographic strategies, which in the current version of the manuscript are exemplified by atomically precise structures of far less complexity than structures formed on alternative systems (see for example ref.2 from the manuscript, or seminal works on CO molecule arrays on Cu surfaces). The latter fact may hinder impact of this work on a larger community.

Main concerns.

The main scientific finding of the manuscript, the repassivation of DBs by STM tip-based strategy is novel, although it follows two recently published articles related to nc-AFM based methods of repassivation (references 5 and 12 respectively). In my opinion the mechanism behind the processes in both cases is of the same origin as Authors do not report any bias dependences in presented data. On the other hand the protocols for atomically precise DB arrays formation are also present in the literature, see for example protocol in ref. 14, which was implemented at RT. In my opinion the stability of the experiments at cryogenic temperatures in this case is also of significant importance. This opens questions how the proposed procedures would work in less strict conditions, for example at RT, as RT stability and use of DB patterns are strongly highlighted in the text.

The manuscript text is well written, however a few important parts of the text may be misleading for the Reader, as the conclusions are not directly supported by the presented data and they are rather extrapolated generalizations. Examples from the abstract:

- 1) (lines 19-21) Authors claim that the hydrogen repassivation strategy is transformed into efficient, accessible and automated error correction/editing tool. However, detailed description in the text states clearly that it is currently not the case and this fact may be realized in the future (lines 205-207)
- 2) (lines 21-23) Stability of large scale atomically precise DB arrays was tested here at cryogenic temperature and only for 72 hours (see lines 140-142).
- 3) (line 23-24) The authors showed rewriting only on the example of 8-bit memory. I have doubts if the procedure is scalable to 192 bits (see my comments below).

As stated above my main concerns are related to scalability of the proposed protocol for hydrogen

removal (lithography). The Fig.1 refers to FT analysis of high resolution empty state STM image, which leads to determination of hydrogen atom sites by the software. However, I have a few questions related to application of this strategy to practical use in an automatized lithography on Si(100):H surface proposed in the manuscript:

1) The clear distinction of atomic sites relies on specific STM apex, which lead to such a contrast in high resolution empty state image of Si:H surface. The Authors do not comment how to solve problems of different STM apexes, which may lead to asymmetric or completely different contrast observed in the same conditions (see for example ref. 14). Are tip reformation protocols also automatized? As I have no doubts that during series of desorption events the exact must be affected. For example STM image in Fig. S2 was obtained in exactly the same conditions as the one in Fig.1 and clearly the contrast on reconstruction rows is different.

2) How the presence of defects on the surface influence the FT analysis? For example in the case of natural atomic terrace edges (seen for example in Fig.2) the Si(100):H reconstruction rows are rotated by 90 degrees.

3) Is the procedure applicable to situation where some of hydrogen sites are already "occupied" by DBs? On the other hand, can one rewrite the already existing pattern with the use of automatized lithographic strategy? In presence of DBs empty state STM images are highly affected by non-trivial effects, which are well known by the Authors (including for example dynamic DBs charging). That affects atomically defined contrast over H sites in empty state STM images around DBs (see for example Fig. S1 or S3).

4) Finally, the size of STM image in Fig.1 is about 2 by 3 nm. Is the procedure applicable for larger scan areas? If yes how large patterns can be prepared automatically by this procedure? Authors do not comment how to surmount fundamental problems related to upscaling of protocols in an automatic manner, like for example STM scanner creep and related drift. Here maybe combination of proposed strategies with the existing methods would be of significant importance (see for example recent product by Scienta Omicron and Zyvex Labs dedicated to nm precise formation of DB patterns).

As regards to hydrogen repassivation in my opinion it is clear that the procedure at current stage is not automatic (as already pointed above), what should be clearly pointed and corrected all along the text. The procedure relies on specific apex conditions, which at current stage cannot be controlled, what results in qualitatively different $I(Z)$ characteristics and more importantly much different experimental conditions during the process (bias voltage alters significantly in the range of $\sim 0.9V$). Moreover, the hydrogen should be loaded on the apex after a few successive events, what in my opinion limits the use of this method for rewriting of the DB based memory consisting of higher number of bits (see my point above).

Some additional suggestions/questions.

1) Line 140-142: "At 4.5 K the memory was unchanged at the end of 72 hours of observation (Supplementary Fig. S5). Held in this environment, samples have shown no significant surface degradation after half a year, so the memory is expected to remain well beyond this period." As pointed in the beginning Authors use often such extrapolated generalizations, what is sharply visible in these particular sentences. In my opinion such a reasoning should be avoided as one can easily find arguments against (Si DBs are highly reactive and even at UHV conditions it is challenging to not passivate them with residual gases during long time intervals) and without a cutoff proof, reasoning based on this type of argumentation is very speculative.

2) In Fig 3 caption I would suggest to add exact temperature of experiment and definition of Z_0 value ($+1.4V$, $50pA?$).

3) In Fig.S3 caption there is a problem with temperature value (should be 77K).

4) In Fig. S4 I would like to ascription of biases to each I(Z) data. Are the currents systematically dependent on biases or different current ranges are rather related to the exact apex structure? It should be noted in the figure caption how is the Z₀ value determined (+1.4V, 50pA?). Can authors estimate how close is the tip to VdV surface contact in positions Z₀ – 550 pm? How the I(Z) curves look like for distances larger than 550pm? Are the trends systematic? Do authors tried to perform similar procedure with low negative biases (in the band gap of n-doped Si:H)?

5) The concept of Figure S5 is not clear for me as it consists two contradictive aspects. The main idea is to show the stability of the structure after 72 hours at LHe conditions. But actually at same the structure was intentionally manipulated, what shows possible rewriting of two bits. I think this should be shown in two separate Figures. Can Author prove the stability by comparison between not transformed structures? I would suggest to remove the word “rewritable” in the case 192 bit memory (see my above comments).

Reviewer #2 (Remarks to the Author):

Report – NCOMMS-18-05289

The manuscript deals with hydrogen removal from and repassivation of dangling bonds (DBs) on hydrogenated Si(100) using the tip of a low-temperature STM. It is shown that reversible hydrogen transfer between Si DB and STM tip enables to create ordered assemblies of up to 62 DBs. The authors argue that this approach represents a leap forward in hydrogen lithography (HL) and eventually in the realization of practical atomic-scale devices. A good review on HL by scanning probe techniques is given and it is concluded that controlled repassivation (error correction) is essential to improve HL capacity. Repassivation by means of hydrogen transfer from an AFM tip to a bare Si DB was reported before by some of the authors (Ref. 12) as well as others (Ref. 5) and attributed to the formation of a tunable silicon-hydrogen covalent bond. In this work, Achal et al. take advantage of the same mechanism by bringing a hydrogen-functionalized STM tip sufficiently close to the DB (denoted type I). An alternative process is described, in which a hydrogen atom from the off-apex region of the tip happens to be transferred (denoted type II). Hydrogen accumulation in the off-apex region was reported before in Ref. 12.

The elementary steps of DB assembly on Si(100) utilized in this work are not new and the functional aspect of the presented structures is limited to their atomic positions (storage). On the other hand, these structures are impressive and the results suggest that significant progress has been made towards robust, reliable, and atomically precise HL fabrication. The latter is the main claim of the work but I'm not yet fully convinced that this is in fact the case. How crucial is the overall tip stability in these experiments when bringing the STM tip close to the surface? How many hydrogen removal/repassivation steps can be done without uncontrolled modification of the tip apex or unintentional dropping of material from the tip? Is the hydrogen accumulation (enabling the type II process) a commonly observed behavior in the experiment or does it rely on an exceptional tip state? The authors should give more information on the reliability of their method. If the authors can provide convincing evidence here, then the work has the right caliber for publication in Nature Communications. The discussion is sound and the quality of the experimental data is very good – no doubt at this point.

Two minor comments:

(1) Literature information of Refs. 9, 26, and 30 needs to be checked.

(2) In the legend of Figure S2 it is stated that “an error has been made” in a hydrogen removal

step. Is it known what the resulting defect or unwanted bonding structure is in this case?

Reviewer #3 (Remarks to the Author):

I have no hesitation in recommending this paper for publication in Nature Communications subject to very minor changes. It is a tour de force demonstration of atomic manipulation, representing a major breakthrough in an intensively studied system: STM-induced hydrogen depassivation. The authors have achieved a step change in our ability to modify matter at the atomic level: the protocols and results they describe show that single chemical bonds can now be controlled in a binary read/write fashion, surmounting a key barrier with the use of H:Si(100) as a substrate for atomic memories. More than this, however, the ability to error correct on a (dangling) bond-by-bond basis opens up exciting possibilities in the generation of cellular automata of the type previously demonstrated by the Wolkow group and, more broadly, a variety of information processing (rather than solely memory) devices.

I recommend publication subject to the following minor changes:

(i) The reference to Michelle Simmons et al.'s work is rather old (2003). I would suggest adding some more recent papers. Please note that I am not a member of Simmons' research team, nor am I an author on any of that group's papers.

(ii) With regard to characterising the state of the probe, did the authors attempt dI/dV vs V spectroscopy to gain insights into changes in density of states? I appreciate just why this could be very difficult, in that the spectroscopy itself could modify the tip apex, but it would be helpful if the authors could add a line or two to the paper to clarify whether they think dI/dV spectroscopy (or other forms of scanning probe 'spectroscopy' such as force-distance curves) might be a viable strategy for tip characterisation.

(iii) I assume that when the authors say they use a 2D Fourier transform, this was just the power spectrum? Or did they analyse the phase too? The latter might be useful in terms of characterising the probe state...

(iv) There are a number of typos, which I assume will be picked up in the proof-reading stage.

I thoroughly enjoyed reading this paper.

Reviewer #4 (Remarks to the Author):

In 'Lithography for Robust, Editable Atomic-scale Silicon Devices and Memories', Roshan Achal and coworkers report on a new STM-based method to repassivate individual dangling bonds on an otherwise hydrogen passivated silicon surface. Hydrogen repassivation (HR) is a highly desired tool to complement the already much more developed hydrogen lithography (HL). Together, the two tools allow for both writing and erasing individual atomic bits. The authors demonstrate this ability by presenting, among others, a 192 bits digital music sample.

The manuscript is well written and the results are convincing and nicely presented. However, I do have some concerns as well as some suggestions for improvement of the paper. I will itemize my issues below.

1. Recently, HR of individual dangling bonds was already demonstrated by means of AFM (Refs. 5 and 12). The authors say about this that "the utility of this technique is limited as AFMs are not ubiquitous and are often more complex to maintain and operate". First, I don't believe that AFMs are any less ubiquitous nor harder to operate than STMs. But even if they were: if it was found

that AFMs are the key to solving all the world's data storage problems, they would of course become ubiquitous soon enough. So this can hardly serve as a valid justification for having to switch to STM.

2. The other method that the current technique is compared to is the storage of data on a chlorine terminated copper surface (Ref. 2). As key difference, the authors state that with their method "the entire surface can be used for storage as the number of available bits is not predetermined at the time of sample preparation". While this may true, they do not mention in their comparison that with the HL/HR method new hydrogen atoms need to be brought in from far away after every few bit flips. In the chlorine vacancy method this is not needed, as the bits there are encoded in the lateral position of a vacancy rather than the presence/absence of a vacancy.

3. Two qualitatively different HR events were recorded (Fig. 3): type I where the tip apex shape alters during HR, and type II where it remains unchanged. The authors suggest that this results from hydrogen atoms being adsorbed respectively to the apex and to the tip side. For type I the current on the downstroke is smaller than on the upstroke, while for type II this is opposite. First, it is not clear to me why the current behaves qualitatively as it does for the two types. Perhaps the authors could provide some clarification here, or at least some speculation. Second, the authors report that they have acquired quite some statistics on the two HR types. My question is: after 'loading' the tip with H-atoms, is the first HR event always type I and all the subsequent events type II? Such information might help in understanding how the tip loading mechanism works and how it might be improved.

4. Related to the above point, I wonder to what extent the findings are truly reproducible. Often in STM, certain manipulation effects may seem very reproducible with one specific microscopic tip shape, but after the tip is deeply indented into the surface or macroscopically altered or replaced altogether, the effects do not return. Were the authors able to built useful memories with significantly different tip shapes? This question is essential in my opinion in view of possible scalability of the technique.

5. The importance of the Fourier analysis method is not quite clear to me. I can understand that such Fourier filtering of the images would help for automated pattern recognition, but is it really a key element in the current finding? The authors devote an entire figure in the main manuscript to it (Fig. 1), which could also be used for some of the analysis suggested above.

In order to make the manuscript suited for publication in Nature Communication, I believe that at least the issues above should be addressed.

Reviewer #1 (Remarks to the Author):

The manuscript “Lithography for Robust, Editable Atomic-scale Silicon Devices and Memories” by Roshan Achal et al. reports on improvements in STM-based lithography technique used for atomically precise formation of dangling-bond (DB) structures on a hydrogen passivated Si(100) surface under UHV conditions and at cryogenic temperatures. Authors introduce a protocol for desorption of single hydrogen atoms from the surface, which can be realized in an automatic manner. Moreover, Authors show novel STM-based method of repassivation of Si dangling bonds by adsorption of single H atoms, which can serve as error-correction or editing tool for the hydrogen lithography. By combination of these two approaches it is demonstrated that several atomically defined patterns of DBs can be formed, with the largest one possessing 62 DBs. Finally, Authors show rewritable pattern of 8 DBs, which serves as 8-bit memory able to store the alphabet letters.

In my opinion the presented results are an important technical step towards upscaling of atomically precise formation of DB patterns on hydrogen passivated semiconductor surfaces. Thus, the work should be of high importance to the community working in this particular field. However, I have doubts if the manuscript will have immediate interest to a broader audience as described protocols are very specific for Si(100):H surface.

We thank the reviewer for highlighting that the work presented is an important step forward in the subject area. We acknowledge that the protocols described here are specific to the Si(100):H system, however, we hope we provided sufficient context in which this system is already of great technological importance to many different communities. We have referenced work in the development of quantum computing devices such as charge qubits, classical computing elements such as single atom transistors and atomic-scale logic gates for instance. The work in quantum computing on Si(100):H with buried dopant atoms, in particular, encompasses some of the most exciting high impact recent results. So, while specific, the protocols here can make an immediate impact in this area of active research, where the ability to create precise sites for the placement of donor atoms is a critical aspect. If the site is incorrectly fabricated, it can now be easily corrected to ensure the intended number of donors are placed to yield the intended device functionality, dramatically reducing experimental variability. Further, with the demonstration of memory functionality we also feel the work overlaps with those interested in high-density storage of data. There are also commercial entities actively exploring/developing technologies based upon these or similar techniques (Quantum Silicon Inc., Zyvex labs, Silicon Quantum Computing). We have added reference to these companies in the introduction to help add additional scope.

A fully general approach to atomic-scale fabrication is beyond the scope of this work, but the focused approach to the Si(100):H system may lay the framework for similar progress on other materials. Hydrogen terminated Si(111) and germanium are natural extensions of this work, where only the parameters of the protocols would have to be adjusted. This would also follow similarly for chlorine terminated Si(100), or hydrogen terminated diamond.

Old Text:

Many disruptive HL-dependant applications have been proposed such as single atom transistors⁶, quantum computing platforms^{7,9}, and atomic-scale logic devices^{4,8}.

New Text:

Many disruptive applications have been proposed based on HL such as single atom transistors⁶, quantum computing platforms^{7,9-11}, and atomic-scale logic devices^{4,8}, drawing both scientific and commercial interest alike. Several companies have even formed upon this and related techniques^{11,14,17}. Overall device development has been delayed, however, by the inability of HL to fabricate large error-free atomic-scale structures^{5,16-21}, increasing the need for reliable error correction techniques.

Old Text:

While these new HL and HR techniques are in their infancy, we demonstrated their immediate utility and applications by creating and editing large error-free DB structures through accessible STM-based means. Both these techniques can be implemented on many STMs with no modifications, operating over a range of temperatures.

New Text:

While these new HL and HR techniques are in their infancy, we demonstrated their immediate utility and applications by creating and editing large error-free DB structures through accessible STM-based means. Both these techniques can be implemented on many STMs with no modifications, operating over a range of temperatures. Though the exact parameters reported here are specific to hydrogen-terminated silicon, they can be adapted to other chemically similar systems such as hydrogen-terminated germanium^{41,42}, hydrogen-terminated diamond^{41,43}, and chlorine-terminated silicon⁴⁴.

Additionally I have concerns about scalability of the presented lithographic strategies, which in the current version of the manuscript are exemplified by atomically precise structures of far less complexity than structures formed on alternative systems (see for example ref.2 from the manuscript, or seminal works on CO molecule arrays on Cu surfaces).

The latter fact may hinder impact of this work on a larger community.

With regards to scalability, we feel this work sets a clear path for the demonstration of much larger and more complex structures. The intention of the work here is to present the first illustration of a combined lithographic approach on silicon that can be easily adopted, allowing for both precise removal of atoms and error correction. The work in reference 2 and with CO both show remarkable control at the atomic-scale, and complex structures. We believe it is now within reach to achieve a similar level of sophistication. We've taken additional steps to show the first automation of tip conditioning, increasing the scalability of the process (now reference 33). In reference 8 we have shown a logic gate structure made of only six dangling bonds, which achieves similar functionality to a significantly more complex logic gate structure made with CO (molecule cascades). The Si system is also qualitatively different than these systems, Si(100):H can be readily integrated within the highly developed semiconductor

fabrication production lines. It is also crucial to keep in mind that the most beautiful recent examples of atom scale patterning were done on systems that are weakly bound and the patterns created do not survive at non-cryogenic conditions. It is far more difficult to break and make bonds controllably when those are of the same approximate strength of the bonds holding the tip and substrate together. The Si(100):H system is stable up to 500 K. Moreover, metal substrates as used previously, do not allow the decoupling of the subtle electronic properties of assembled surface structures from those of the substrate in a way likely to be useful for electronic circuitry. So, for these reasons we feel the system of study that is our focus is uniquely interesting to the larger community.

Main concerns.

The main scientific finding of the manuscript, the repassivation of DBs by STM tip-based strategy is novel, although it follows two recently published articles related to nc-AFM based methods of repassivation (references 5 and 12 respectively). In my opinion the mechanism behind the processes in both cases is of the same origin as Authors do not report any bias dependences in presented data. On the other hand the protocols for atomically precise DB arrays formation are also present in the literature, see for example protocol in ref. 14, which was implemented at RT. In my opinion the stability of the experiments at cryogenic temperatures in this case is also of significant importance. This opens questions how the proposed procedures would work in less strict conditions, for example at RT, as RT stability and use of DB patters are strongly highlighted in the text.

We thank the reviewer for bringing this point to our attention. Hydrogen lithography, as they pointed out, has been demonstrated at room temperature in reference 14 (of the original document). Additionally, it has been shown in several of the other references, providing sizable evidence that the lithography aspect itself would work in less strict conditions than presented in the manuscript. Based on these results, and our experience, the pulse duration and amplitude in our protocols would only need minor adjustments when working above cryogenic conditions. We have not yet had the opportunity to test the repassivation protocols at room temperature, however, we have shown that the procedure works up to 77 K. There is evidence to suggest that it will also work at room temperature, as Yamamoto et al. (reference 30) demonstrated the existence of atomic H on the tip at room temperature, which is the critical aspect of the protocol. Additionally, uncontrolled transfer from the STM tip to a DB at room temperature has been reported in reference 39.

The reviewer's question has highlighted a lack of clarity in our manuscript, as the stability of the structures and the conditions under which they are fabricated are not necessarily related. It is conceivable to fabricate structures at low temperatures for later use at room temperature. Fabrication of DB structures at room temperature for use at room temperature would be ideal but is not the claim of this work. That said, we have developed strategies to work at warmer temperatures, should the need arise, that are now discussed more directly in the main manuscript. Working at elevated temperatures increases the need to address thermal drift and creep for improved accuracy during HL.

Old Text:

NA

New Text:

An important consideration inherent in all scanned probe lithography is the existence of thermal drift and creep, both of which can also cause uncertainty in the position of the tip, leading to errors. At 4.5 K these factors can be well controlled by allowing the STM to stabilize over a period of several hours. However, at warmer temperatures or in situations where allowing the STM to stabilize is not an option, a more active solution is required. To address these factors, we implemented periodic image realignments into the HL workflow. Before initiating the HL procedure, an area near the lithography location ($\sim 10 \times 10 \text{ nm}^2$) is imaged as a reference. After a set time, lithography is paused, and this area is reimaged to determine how much the tip has been offset from its intended position due to creep and drift. The remaining sites in the pattern are shifted appropriately to compensate and lithography resumes. The effectiveness of this realignment can be increased by reducing the interval between reference checks, permitting an optimization between speed and accuracy depending on a given application. We found that without realignment the lithographic accuracy during HL using a non-stabilized STM was near 35% for a particular structure. Under the same conditions using moderate active realignment it was over 85%, which is within a suitable range to then correct the remaining errors using HR.

Old Text:

While these new HL and HR techniques are in their infancy, we demonstrated their immediate utility and applications by creating and editing large error-free DB structures through accessible STM-based means. Both these techniques can be implemented on many STMs with no modifications, operating over a range of temperatures.

New Text:

While these new HL and HR techniques are in their infancy, we demonstrated their immediate utility and applications by creating and editing large error-free DB structures through accessible STM-based means. Both these techniques can be implemented on many STMs with no modifications, operating over a range of temperatures. Though the exact parameters reported here are specific to hydrogen-terminated silicon, they can be adapted to other chemically similar systems such as hydrogen-terminated germanium^{41,42}, hydrogen-terminated diamond^{41,43}, and chlorine-terminated silicon⁴⁴. There is also the potential to eventually extend these protocols to room temperature, as forms of HL have already been successfully demonstrated there^{16,20,23,26}, and the uncontrolled transfer of hydrogen from an STM tip to a DB has been observed for HR³⁹. However, fabricating structures/devices at cryogenic temperatures (where creep and thermal drift are not as pronounced) for use at room-temperature may already be sufficient for many applications. Further, the high-temperature stability of DBs removes one of the logistical issues surrounding the transportation of fabricated nano-devices to an end user, regardless of the temperature required for their operation.

The manuscript text is well written, however a few important parts of the text may be misleading for the Reader, as the conclusions are not directly supported by the presented data and they are rather extrapolated generalizations. Examples from the abstract:

1) (lines 19-21) Authors claim that the hydrogen repassivation strategy is transformed into efficient, accessible and automated error correction/editing tool. However, detailed description in the text states clearly that it is currently not the case and this fact may be realized in the future (lines 205-207)

We see how this wording may lead to confusion. The repassivation strategy we present is both efficient and accessible compared to any existing protocols available, allowing us to now create the presented structures with relative ease. We did not intend to suggest that the procedure is mature enough to be production ready, as only the protocols after a site for repassivation has been selected have been automated. The word automated has been removed from the abstract in reference to HR to better reflect this.

Old Text:

Here, we report scanning tunneling microscope (STM) techniques to substantially improve automated HL and to transform state-of-the-art hydrogen repassivation (HR)^{5,12} into an efficient, accessible, automated error correction/editing tool.

New Text:

Here, we report scanning tunneling microscope (STM) techniques to substantially improve automated hydrogen lithography (HL) on silicon, and to transform state-of-the-art hydrogen repassivation (HR) into an efficient, accessible error correction/editing tool relative to existing chemical and mechanical methods.

2) (lines 21-23) Stability of large scale atomically precise DB arrays was tested here at cryogenic temperature and only for 72 hours (see lines 140-142).

We mention the ability to create room temperature stable structures based on a body of work detailing the stability of dangling bonds. Based on the experimentally determined thermal activation barriers for dangling bonds to hop, it is not an extrapolation in our opinion that any structure created in cryogenic conditions will survive at room temperature. This is further supported by the creation of dangling bond structures at room temperature by ourselves and other groups. The observation of the DB array over 72 hours served to emphasize this point, as many other systems will show disordering after such a time period even at cryogenic conditions. Further, it was also to illustrate a resilience to the pressure and temperature spikes that occurred when the STM cryostat was filled with liquid helium. We have added the following discussion to the manuscript to better substantiate claims and reduce speculations.

Old Text:

At 4.5 K the memory was unchanged at the end of 72 hours of observation (Supplementary Fig. S5). Held in this environment, samples have shown no significant surface degradation after half a year, so the memory is expected to remain well beyond this period. While the room-temperature stability of the memory could not be demonstrated here, room-temperature stable DB structures have already been established in prior works^{4,15,16,24,25}. DBs on the surface of silicon are known to face barriers to diffusion in excess of 1.4 eV in either direction^{10,11}. Such stability and density make DB-based memories a uniquely equipped candidate for data archival and long term storage.

New Text:

We can read the 192-bit memory directly from the STM as the tip scans over the structure, or later from recorded images (see Supplementary movie). The room-temperature stability of DB structures has already been well established in prior works^{4,17,18,26,28}. DBs on the surface of silicon are known to face barriers to diffusion in excess of 1.4 eV in either direction^{12,13}. Taking an exponential prefactor of $A = 2 \cdot 10^{14} \text{ s}^{-1}$ and the lowest energy barrier of 1.46 eV¹³ (intradimer diffusion), a DB is expected to experience only one hop in over 500 years due to thermally activated processes at 300 K. Such stability and density make DB-based memories a unique candidate for data archival and long-term storage.

Held at 4.5 K we detected no unintentional changes in the memory at the end of 72 hours of observation (Supplementary Fig. S5). In this environment, we have worked with samples that have shown no significant surface degradation after half a year. The ultra-high vacuum requirements may also eventually be relaxed as isolated DBs can be protected against spontaneous reactions through the appropriate choice of doping level of the silicon substrate. Highly doped n-type silicon results in negatively charged DBs, which have a barrier to reaction with closed shell species³⁷. There is also only a subset of entities that are known to readily react with DBs^{27,37-39}. Molecular hydrogen, which is commonly present in vacuum environments, requires two directly adjacent DBs along a dimer row for adsorption⁴⁰, reducing the likelihood of spontaneous repassivation of isolated DBs at greater separations (as in the memory) due to ambient gas. The inability to eliminate all naturally occurring DBs during sample preparation in an environment of $1 \cdot 10^{-6}$ Torr of hydrogen gas, or with intentional chemical dosing further supports this notion²⁷.

3) (line 23-24) The authors showed rewriting only on the example of 8-bit memory. I have doubts if the procedure is scalable to 192 bits (see my comments below).

We hope to convince the reviewer that any DB structure can be edited/rewritten using the HR techniques. The larger memory presented here can be thought of as 24 8-bit memories, each separated by 0.768 nm. The ability to rewrite the 192-bit memory then extends from our demonstration of rewriting the 8-bit memory in figure 4a. We have also shown in figure 2, figure 4a, and figure S1 that we can readily address atomic sites spaced closer than 0.768 nm without altering the surrounding structures. In figure S5a the original structure was created correctly according to the design input into the program. However, due to human error the design itself was incorrect, so a line of the memory had to be rewritten (figure S5b) in order to achieve the intended design/structure, as discussed by the reviewer below. This demonstration of rewriting data within the 192-bit memory serves as an example of the techniques used in the 8-bit memory applying to the larger memory.

As stated above my main concerns are related to scalability of the proposed protocol for hydrogen removal (lithography). The Fig.1 refers to FT analysis of high resolution empty state STM image, which leads to determination of hydrogen atom sites by the software. However, I have a few questions related to application of this strategy to practical use in an automatized lithography on Si(100):H surface proposed in the manuscript:

1) The clear distinction of atomic sites relies on specific STM apex, which lead to such a contrast in high resolution empty state image of Si:H surface. The Authors do not comment how to solve problems of different STM apexes, which may lead to asymmetric or completely different contrast observed in the same conditions (see for example ref. 14). Are tip reformation protocols also automatized? As I have no doubts that during series of desorption events the exact must be affected. For example STM image in Fig. S2 was obtained in exactly the same conditions as the one in Fig.1 and clearly the contrast on reconstruction rows is different.

We thank the reviewer for bringing up this important point for further discussion. So long as the tip is in good condition, we can reliably perform all the protocols described. We have been developing an automated tip forming protocol, which is capable of determining when a tip is of the necessary quality (ref 33). More discussion to this point has now been added into the text.

Old Text:

NA

New Text:

Using this procedure, the probability of detrimental uncontrolled apex changes is low. By beginning removal attempts at 1.8 V (see Methods), higher voltages, which are more likely to change or damage the tip, are only reached when necessary. Conservatively, on the order of 10 DBs can be created consecutively without some type of minor modification to the tip. However, we have found that HL efficiency is not particularly sensitive to minor changes of the tip, so the actual number of DBs that can be created without altering removal efficiency during fabrication is often larger. Should the tip change so much that it is no longer suitable for HL purposes, an automated tip forming routine can be called to recondition the tip through controlled contact with the surface³³. This routine takes advantage of a machine learning algorithm, and STM image data for training sets, to automatically identify the quality of the probe by imaging a DB, initiating reconditioning when necessary³³.

2) How the presence of defects on the surface influence the FT analysis? For example in the case of natural atomic terrace edges (seen for example in Fig.2) the Si(100):H reconstruction rows are rotated by 90 degrees.

Localized defects (including step edges, which are localized in one dimension), by definition contribute to many different frequency components, while the extended periodicity of the surface leads to a sharp peak in the FT (figure 1b). Such features in the FT are thus quite robust, even in the presence of imperfections. When images contain multiple terraces, the FT no longer reports a 2-fold symmetry, as shown in figure 1b, but appears to show a 4-fold symmetry. The procedure can still be applied, although one must choose the correct periodicities for each terrace. Generally, the fabrication and consequently the image analysis is currently only performed on one terrace. Related text has been added to the manuscript (see point #4 below).

3) Is the procedure applicable to situation where some of hydrogen sites are already “occupied” by DBs?

Yes, the procedure is applicable in this situation. When the dangling bonds represent a small portion of the overall image used in the procedure, then as discussed above in question 2, the FT is quite robust to their presence. Should the presence of unwanted DBs begin to affect the proper characterization, HR allows us to condition the area by removing them before lithography.

On the other hand, can one rewrite the already existing pattern with the use of automatized lithographic strategy?

It is possible to write to an existing pattern using the automatic lithography strategies. From figure 2c to figure 2d the sites identified at the beginning of the HL procedure were used to correctly place the remaining two dangling bonds within the existing structure.

In presence of DBs empty state STM images are highly affected by non-trivial effects, which are well known by the Authors (including for example dynamic DBs charging). That affects atomically defined contrast over H sites in empty state STM images around DBs (see for example Fig. S1 or S3).

The FT is also robust in the situation of the altered contrast near DBs, as the periodicity of the surface tends to extend well beyond the area altered by the DB charging effects. For example, as in figure S1b.

4) Finally, the size of STM image in Fig.1 is about 2 by 3 nm. Is the procedure applicable for larger scan areas?

The procedure is generally applied to larger scan areas. We see it was not clear that the selection of figure 1 was intended for illustrative purposes without the added complication of defects in the discussion. Typical image size for the automated method is between 10 nm x 10 nm to 40 nm x 40 nm.

In the case of figure 2 the leaf was constructed using an image of 12 nm x 12 nm. Further discussion has been added into the text to more clearly indicate these aspects.

Old Text:

To begin automated HL, the location of every hydrogen atom in a select area is determined for accurate STM tip registration during fabrication. The periodicity of the hydrogen passivated Si(100)-2x1 surface (Fig. 1a) permits the location of every hydrogen atom to be determined from a single STM image (while accounting for nonlinearities in the scanner) through the use of Fourier analysis²⁸ (Fig. 1b-f).

New Text:

To begin automated HL, the location of every hydrogen atom in a select area needs to be determined for accurate STM tip registration during fabrication. Slight errors in the tip position can result in incorrect atoms being removed. Fast, fully autonomous lithography also requires the location of each atom to be known, such that the surface doesn't need to be reimaged after each removal event to determine the next site. The periodicity of the hydrogen-passivated Si(100)-2x1 surface (Fig. 1a) permits the location of every hydrogen atom to be determined from a single STM image (while accounting for nonlinearities in the scanner) through the use of Fourier analysis³¹ (Fig. 1b-f). Such an analysis is relatively immune to the presence of small surface defects and dangling bonds due to their spatially localized nature in the images compared to the extended periodicity of the surface itself. Figure 1 illustrates the basic features of this process. In practice, we use images between $10 \times 10 \text{ nm}^2$ to $40 \times 40 \text{ nm}^2$ to determine the location of the hydrogen atoms on a given sample terrace.

If yes how large patterns can be prepared automatically by this procedure?

In principle, if HL was 100% efficient, using the simplest implementation of this protocol, patterns as large as a given terrace could be created through automated means. However, such attempts have not been actively explored since HL efficiency was not sufficient. Now with error correction at hand, much larger patterns will be possible, as we have begun to demonstrate with the structures in this work.

Authors do not comment how to surmount fundamental problems related to upscaling of protocols in an automatic manner, like for example STM scanner creep and related drift. Here maybe combination of proposed strategies with the existing methods would be of significant importance (see for example recent product by Scienta Omicron and Zyvex Labs dedicated to nm precise formation of DB patterns).

These are important considerations for scaling production, and we are glad to discuss them further. Strategies to surmount drift and creep have been presented in our response above question 1, with additional material added into the manuscript (shown here again). Should these strategies eventually prove insufficient as the system scales, then the reviewers suggestion of a combined approach with those of Zyvex/Omicron presents another very viable solution.

Old Text:

NA

New Text:

An important consideration inherent in all scanned probe lithography is the existence of thermal drift and creep, both of which can also cause uncertainty in the position of the tip, leading to errors. At 4.5 K these factors can be well controlled by allowing the STM to stabilize over a period of several hours. However, at warmer temperatures or in situations where allowing the STM to stabilize is not an option, a more active solution is required. To address these factors, we implemented periodic image realignments into the HL workflow. Before initiating the HL procedure, an area near the lithography location ($\sim 10 \times 10 \text{ nm}^2$) is imaged as a reference. After a set time, lithography is paused, and this area is reimaged to determine how much the tip has been offset from its intended position due to creep and drift. The remaining sites in the pattern are shifted appropriately to compensate and lithography resumes. The effectiveness of this realignment can be increased by reducing the interval between reference checks, permitting an optimization between speed and accuracy depending on a given application. We found that without realignment the lithographic accuracy during HL using a non-stabilized STM was near 35% for a particular structure. Under the same conditions using moderate active realignment it was over 85%, which is within a suitable range to then correct the remaining errors using HR.

As regards to hydrogen repassivation in my opinion it is clear that the procedure at current stage is not automatic (as already pointed above), what should be clearly pointed and corrected all along the text.

We can see where the confusion arises in the manuscript. The repassivation procedure as described in the main text has been automated, the only user intervention required is to select the location for HR. The steps such as changing bias, sweeping height, recording tunneling current, and resetting position are completed at the push of a button without user intervention. The word “automated” has been removed from the abstract in reference to HR. We discuss a strategy for eventual fully integrated repassivation within the automated lithography workflow, and more clearly indicate what has been automated in various places in the text. The major changes are shown below.

Old Text:

With a functionalized tip, the first step of HR is to position it over a DB at a sample voltage of 1.4 V and current of 50 pA. The feedback control is then disabled and the sample voltage is changed to a value between 100 mV to 1.0 V. While recording the tunneling current, the STM tip is brought 500 to 800 pm towards the sample, then is retracted to its original position. The voltage is reset to the original value of 1.4 V and the feedback control is restored. The entire process takes ~ 1 s and can be repeated until a successful repassivation signature is observed.

New Text:

With a functionalized tip, the first step of HR is to position it over a DB at a sample voltage of 1.4 V and current of 50 pA. The feedback control is then disabled, and the sample voltage is changed to a value between 100 mV and 1.0 V. While recording the tunneling current, the STM tip is brought 500 to 800 pm towards the sample, then is retracted to its original position. The voltage is reset to the original value of 1.4 V and the feedback control is restored. This entire process, once a user has selected a site, has been automated, taking ~1 s. It can be initiated at the press of a button and repeated until a successful repassivation signature is observed. Work is underway to integrate this new HR process within the HL workflow to enable fully autonomous fabrication and correction. Errors will be automatically detected via image recognition, and subsequently corrected using the HR technique (see Methods).

Old Text:

The tip is set directly over the lattice site where a DB is present ($I=50$ pA, $V=1.4$ V). The feedback controls are switched off, locking the tip-height. The sample voltage is changed to a value between 100 mV to 1.0 V, and then the tip is moved linearly towards the sample surface while recording the tunneling current. After the tip has traveled a distance of 550 pm towards the surface it is retracted to its original position. The original parameters are re-established, and the feedback control is restored. To date, no significant correlation between voltage and HR efficiency has been observed. The choice in voltage serves to limit the tunneling current to within ranges that prevent significant tip apex changes, while still providing adequate feedback signals. If the initial HR attempt is unsuccessful, the process can be repeated until a type-I or type-II signature is detected. Work is in progress to include automatic error detection after HL and to define arbitrary groups of sites for HR in an image. This will eventually enable fully automated HR without the need for users to select individual sites to initiate the HR process.

New Text:

The tip is set directly over the lattice site where a DB is present ($I=50$ pA, $V=1.4$ V). The feedback controls are switched off, locking the tip-height. The sample voltage is changed to a value between 100 mV and 1.0 V, and then the tip is moved linearly towards the sample surface while recording the tunneling current. After the tip has traveled a distance of 550 pm towards the surface it is retracted to its original position. The original parameters are re-established, and the feedback control is restored. To date, no significant correlation between voltage and HR efficiency has been observed. The choice of voltage serves to limit the tunneling current to within ranges that prevent significant tip apex changes, while still providing adequate feedback signals. Typically, we perform HR at a bias of 200 mV and only adjust this value in the program when the signal falls outside of the desired range (3 pA to 300 pA). The strength of the signatures depends on the exact structure of the apex, as they can vary by an order of magnitude at the same applied bias (Supplementary Fig. S3, S4). Even though the strength of the signatures vary, their shape remains characteristic, making them ideal for the detection of successful events (Supplementary Fig. S3, S4). If the initial HR attempt is unsuccessful, the process can be repeated until a type-I or type-II signature is detected. Work is in progress to include automatic error detection after HL using image recognition to define arbitrary groups of sites for HR in an image. This will eventually enable fully automated HR, without any user intervention to select individual sites to initiate the HR process.

The procedure relies on specific apex conditions, which at current stage cannot be controlled, what results in qualitatively different I(Z) characteristics and more importantly much different experimental conditions during the process (bias voltage alters significantly in the range of ~0.9V).

The apex conditions, as discussed with automated tip forming, can be controlled to a reasonable degree. In the case of type-II repassivation events, it appears the apex structure is not highly specific as many tips have shown such functionality, even after tip forming resulting in quantitatively different imaging resolution. Figure 3c consists of data from a total of seven physically different tips, suggesting the conditions for repassivation are more general. If it is eventually found that a particular tip apex is needed, it will be possible to more exactly select an appropriate apex with sufficient training data in the tip forming program to reduce the variability in experimental parameters. We note the possible confusion from the appropriate bias ranges used. We wanted to present the most general description of our observations, and where the protocols work. In practice, we have the bias set at 0.2 V and only adjust it if we get signal outside of the range of 3 pA – 300 pA, which is not often the case. The text has been refined for clarity of this aspect (see above).

Moreover, the hydrogen should be loaded on the apex after a few successive events, what in my opinion limits the use of this method for rewriting of the DB based memory consisting of higher number of bits (see my point above).

For larger patterns, as hydrogen is removed, there will be more H available on the tip for repassivation. The need to reload the tip is often a consequence of the low number of DBs used in the designs we have begun working with. We have modified the text to reflect this. Other strategies to improve the rewriting speeds and practicality of DB-based memories, discussed in the text, include the possible use of different tip materials. Reference 29 has shown that Pt tips are capable of holding ~1000 atoms.

Old Text:

The HR stage is currently the slowest step, limited by the number of available hydrogen atoms on the surface of the tip. Moving off of the structure to reload the tip after repassivating several DBs introduced a significant delay. Improvements to HR speeds are possible through enhanced automation and through tip materials like platinum²⁶ capable of holding more hydrogen.

New Text:

The HR stage is currently the slowest step, limited by the number of available hydrogen atoms on the surface of the tip. Moving away from the structure to reload the tip after repassivating several DBs introduced a significant delay. This may only be a factor for structures with a small number of DBs, like those we have presented. With structures requiring more DBs there will be a continued source of hydrogen to the tip as each new DB is created (equivalent to the reloading procedure). With enhanced automation to incorporate periodic intervals for HR/error correction during HL the need to travel away from the structure to reload the tip can be reduced or altogether eliminated. Further improvements to HR speeds may be possible through tip materials like platinum, which is able to hold at least 1000 atoms of hydrogen on its surface²⁹.

Some additional suggestions/questions.

1) Line 140-142: "At 4.5 K the memory was unchanged at the end of 72 hours of observation (Supplementary Fig. S5). Held in this environment, samples have shown no significant surface degradation after half a year, so the memory is expected to remain well beyond this period." As pointed in the beginning Authors use often such extrapolated generalizations, what is sharply visible in these particular sentences. In my opinion such a reasoning should be avoided as one can easily find arguments against (Si DBs are highly reactive and even at UHV conditions it is challenging to not passivate them with residual gases during long time intervals) and without a cutoff proof, reasoning based on this type of argumentation is very speculative.

Additional details have been added to the main text and methods sections to address these concerns and reduce speculations. More information has been added to discuss the reactivity of DBs as well, which as the reviewer pointed out is an important consideration. DB reactivity has now been better studied for many species, and only a handful have been shown to react with isolated DBs (references 27,37,38,39). We have also attempted to reduce the concentration of naturally occurring DBs during sample preparation by leaving the sample exposed to hydrogen gas for an extended period before transferring it into the scanning chamber. Such attempts have not resulted in a quantifiable change in DB concentration.

Old Text:

Held in this environment, samples have shown no significant surface degradation after half a year, so the memory is expected to remain well beyond this period. While the room-temperature stability of the memory could not be demonstrated here, room-temperature stable DB structures have already been established in prior works^{4,15,16,24,25}. DBs on the surface of silicon are known to face barriers to diffusion in excess of 1.4 eV in either direction^{10,11}. Such stability and density make DB-based memories a uniquely equipped candidate for data archival and long term storage.

New Text:

Held at 4.5 K we detected no unintentional changes in the memory at the end of 72 hours of observation (Supplementary Fig. S5). In this environment, we have worked with samples that have shown no significant surface degradation after half a year. The ultra-high vacuum requirements may also eventually be relaxed as isolated DBs can be protected against spontaneous reactions through the appropriate choice of doping level of the silicon substrate. Highly doped n-type silicon results in negatively charged DBs, which have a barrier to reaction with closed shell species³⁷. There is also only a subset of entities that are known to readily react with DBs^{27,37-39}. Molecular hydrogen, which is commonly present in vacuum environments, requires two directly adjacent DBs along a dimer row for adsorption⁴⁰, reducing the likelihood of spontaneous repassivation of isolated DBs at greater separations (as in the memory) due to ambient gas. The inability to eliminate all naturally occurring DBs during sample preparation in an environment of $1 \cdot 10^{-6}$ Torr of hydrogen gas, or with intentional chemical dosing further supports this notion²⁷.

Old Text:

Following that, we exposed the sample to $1 \cdot 10^{-6}$ Torr of hydrogen gas. A nearby tungsten filament held at $1900 \text{ }^\circ\text{C}$ was used to crack the gas into atomic hydrogen. We exposed the sample to the gas for 120 s with no heating, then rapidly flashed it to $1250 \text{ }^\circ\text{C}$, after which we quickly brought the temperature down to $330 \text{ }^\circ\text{C}$ for 150 s to achieve the desired hydrogen terminated 2×1 surface reconstruction.

New Text:

Following that, we exposed the sample to $1 \cdot 10^{-6}$ Torr of hydrogen gas. A nearby tungsten filament held at $1900 \text{ }^\circ\text{C}$ was used to crack the gas into atomic hydrogen. We exposed the sample to the gas for 120 s with no heating, then rapidly flashed it to $1250 \text{ }^\circ\text{C}$, after which we quickly brought the temperature down to $330 \text{ }^\circ\text{C}$ for 150 s to achieve the desired hydrogen-terminated 2×1 surface reconstruction. The sample remains in the preparation chamber for up to 15 minutes as the pressure slowly returns towards the initial base pressure.

2) In Fig 3 caption I would suggest to add exact temperature of experiment and definition of Z₀ value ((+1.4V, 50pA?).

These details have been added.

a, ($V=0.4 \text{ V}$, $T=4.5 \text{ K}$) The recorded tunneling current as the STM tip (set over a DB at 1.4 V and 50 pA) is brought towards the surface (blue) and as the STM tip is retracted (red) during HR.

3) In Fig.S3 caption there is a problem with temperature value (should be 77K).

Thank you for identifying this error. The text has been corrected.

a, ($V=0.2 \text{ V}$) Type-II signature in the STM tunneling current recorded during HR at 77 K . **b-c**, ($V=1.4 \text{ V}$, $I=50 \text{ pA}$, $T=77 \text{ K}$, $4.5 \times 9.3 \text{ nm}^2$) STM images before and after successful HR.

4) In Fig. S4 I would like to ascription of biases to each I(Z) data.

A bias value has been added for each plot. (See below).

Are the currents systematically dependent on biases or different current ranges are rather related to the exact apex structure? It should be noted in the figure caption how is the Z_0 value determined (+1.4V, 50pA?).

The apex structure appears to be the main differentiating factor for the different current ranges observed. With a bias of 0.2 V, the observed currents can differ significantly within the range of 3 pA to 300 pA.

Old Text:

a-b, A representative sample of different type-I and type-II signatures recorded during HR events. Applied bias voltages were in between 0.2 and 0.7 V. While the magnitude of both signatures can vary depending on the applied voltage during HR and changes in apex orbital, their overall shapes remain very characteristic. The reliability and reproducibility of these features makes them excellent indicators of successful repassivation for automation routines.

New Text:

a-b, A representative sample of different type-I and type-II signatures recorded during HR events ($T=4.5$ K), with the tip set over a DB at 1.4 V and 50 pA. Applied bias voltages in **a**, from left to right, are 0.5 V, 0.3 V, 0.2 V, 0.2 V respectively. Applied bias voltages in **b**, from left to right, are 0.2 V, 0.4 V, 0.2 V, 0.2 V respectively. While the magnitude of both signatures can vary depending on the apex orbital and the choice of applied voltage to limit the tunneling current during HR, their overall shapes remain very characteristic. The reliability and reproducibility of these features makes them excellent indicators of successful repassivation for automation routines.

Can authors estimate how close is the tip to VdV surface contact in positions $Z_0 - 550$ pm?

Set over a dangling bond at 1.4 V, 50 pA we have observed that changes to the surface begin to occur with an approach distance of 900 pm under low applied biases ($V=0.2$ V). We then estimate that $Z_0 = \sim 900$ pm, so $Z_0 - 550$ pm = ~ 350 pm.

How the I(Z) curves look like for distances larger than 550pm?

Beyond 550 pm towards the surface, the curves often remain exponential, for example:

At closer approach distances (greater than ~ 850 pm) the curves may deviate, likely due to induced changes in the apex or surface with such close proximity to the surface.

Are the trends systematic? Do authors tried to perform similar procedure with low negative biases (in the band gap of n-doped Si:H)?

We have explored this region in a limited way. Preliminary results have shown several successful repassivation events at small negative biases. It could be an area of further study.

5) The concept of Figure S5 is not clear for me as it consists two contradictive aspects. The main idea is to show the stability of the structure after 72 hours at LHe conditions. But actually at same the structure was intentionally manipulated, what shows possible rewriting of two bits. I think this should be shown in two separate Figures. Can Author prove the stability by comparison between not transformed structures? I would suggest to remove the word “rewritable” in the case 192 bit memory (see my above comments).

We agree with the reviewer, that under ideal circumstances it would be better if we had an untransformed structure, we do not routinely observe static structures for long periods of time, as experiments often require changing many aspects of it, or the surrounding area. The purpose of figure S5 was to illustrate three aspects of the memory. The first aspect was the lack of thermal diffusion, aside from the one dangling bond intentionally rewritten, the other 61 did not move over the time we observed the structure. The second objective was to show that no DB reacted with ambient gases in vacuum even during the pressure and temperature spikes associated with filling the machine. The third aspect was to show a degree of rewritability of the memory, as discussed earlier, where two bits had to be rewritten due to an input error into the lithography program. We feel this set of images achieved all three goals that it was intended to show. As a further example, here is a 35 nm x 35 nm image of the hydrogen terminated surface with an arbitrary geometric structure created from DBs after 72 h of observation ($T= 4.5$ K, $V=-1.8$ V, $I=50$ pA). No movement of the DBs present in the image can be

observed, nor surface contamination resulting from the increased pressure and temperature encountered while filling the machine with cryo-fluids as described in the figure caption. This is also the case for the 90 nm x 90 nm image (T= 4.5 K, V=-1.8 V, I=50 pA) shown below after 188 hours of observation. The tip was sharpened during this time interval, causing the DBs in the second image to look narrower.

Reviewer #2 (Remarks to the Author):

Report – NCOMMS-18-05289

The manuscript deals with hydrogen removal from and repassivation of dangling bonds (DBs) on hydrogenated Si(100) using the tip of a low-temperature STM. It is shown that reversible hydrogen transfer between Si DB and STM tip enables to create ordered assemblies of up to 62 DBs. The authors argue that this approach represents a leap forward in hydrogen lithography (HL) and eventually in the realization of practical atomic-scale devices. A good review on HL by scanning probe techniques is given and it is concluded that controlled repassivation (error correction) is essential to improve HL capacity. Repassivation by means of hydrogen transfer from an AFM tip to a bare Si DB was reported before by some of the authors (Ref. 12) as well as others (Ref. 5) and attributed to the formation of a tunable silicon-hydrogen covalent bond. In this work, Achal et al. take advantage of the same mechanism by bringing a hydrogen-functionalized STM tip sufficiently close to the DB (denoted type I).

An alternative process is described, in which a hydrogen atom from the off-apex region of the tip happens to be transferred (denoted type II). Hydrogen accumulation in the off-apex region was reported before in Ref. 12.

The elementary steps of DB assembly on Si(100) utilized in this work are not new and the functional aspect of the presented structures is limited to their atomic positions (storage). On the other hand, these structures are impressive and the results suggest that significant progress has been made towards robust, reliable, and atomically precise HL fabrication. The latter is the main claim of the work but I'm not yet fully convinced that this is in fact the case.

How crucial is the overall tip stability in these experiments when bringing the STM tip close to the surface?

As in many scanned probe experiments, the overall stability of the tip is important to ensure reliable and reproducible events. The tip cleaning and preparation procedures we describe goes a long way to ensuring a clean, stable, and relatively predictable tip structure. The parameters we have presented in this work are gentle in the sense that we very seldom observe random uncontrolled apex changes during a repassivation event. Such structural changes are induced by sudden high currents, or uncontrolled crashes into the sample surface. By selecting a maximum distance of 550 pm for an approach towards the surface, we reduce the likelihood of a crash, as the tip is still at least 300 pm from the surface. Additionally, the choice of low bias voltages helps ensure that large currents capable of altering the structure of the tip are avoided.

While tip stability is important, it is also true that the procedures we describe can be carried out with most "good" tips. Some tip changes can thus be tolerated, and we can quite reliably recover a tip by making controlled contact with the surface. The work in reference 33 describes a method to automate tip forming, which will enable unstable tips to be detected and corrected via controlled contact with the surface. We have added some discussion into the methods section.

Old Text:

The choice in voltage serves to limit the tunneling current to within ranges that prevent significant tip apex changes, while still providing adequate feedback signals.

New Text:

The choice of voltage serves to limit the tunneling current to within ranges that prevent significant tip apex changes, while still providing adequate feedback signals. Typically, we perform HR at a bias of 200 mV and only adjust this value in the program when the signal falls outside of the desired range (3 pA to 300 pA).

How many hydrogen removal/repassivation steps can be done without uncontrolled modification of the tip apex or unintentional dropping of material from the tip?

Conservatively estimating, approximately 10 hydrogen removal events can be done without minor uncontrolled modification of the tip. Often this number is higher in practice, as the attempts at lower voltages in the automated routine help reduce the likelihood of random tip changes due to large voltage pulses. The lithography procedure has been found to be robust to minor uncontrolled changes that occur during extended lithography as well, where significantly more than 10 atoms are successfully removed without checking the tip for changes via an STM image. In our experience creating the structures presented here, along with other work, so long as the tip has not had an uncontrolled crash, dropping of material from the tip has not been observed during either an HL or HR event. Text has been added to the manuscript discussing how uncontrolled changes can be overcome. In terms of repassivation, aside from the apex atom of the tip changing during a type-I event, we have not observed uncontrolled modification of the tip, unless the current reaches values in excess of 500 pA. Even at such currents, the apex will not necessarily change. The low probability of uncontrolled modification during an ideal HR event, as discussed in the previous point is largely due to the low currents involved, along with maintaining a safe tip-sample separation.

Old Text:

NA

New Text:

Using this procedure, the probability of detrimental uncontrolled apex changes is low. By beginning removal attempts at 1.8 V (see Methods), higher voltages, which are more likely to change or damage the tip, are only reached when necessary. Conservatively, on the order of 10 DBs can be created consecutively without some type of minor modification to the tip. However, we have found that HL efficiency is not particularly sensitive to minor changes of the tip, so the actual number of DBs that can be created without altering removal efficiency during fabrication is often larger. Should the tip change so much that it is no longer suitable for HL purposes, an automated tip forming routine can be called to recondition the tip through controlled contact with the surface³³. This routine takes advantage of a

machine learning algorithm, and STM image data for training sets, to automatically identify the quality of the probe by imaging a DB, initiating reconditioning when necessary³³.

Is the hydrogen accumulation (enabling the type II process) a commonly observed behavior in the experiment or does it rely on an exceptional tip state? The authors should give more information on the reliability of their method.

This is an excellent question, we thank the reviewer for prompting further discussion. We have not found the type-II repassivation process to require an exceptional tip state. The repassivation data in figure 3c was collected using 7 physically different tips. The ability to achieve a type-II repassivation persists even after moving the tip 2 nm towards the surface, drastically altering its structure (~1 nm beyond initial contact with the surface). Additional details have been added within the manuscript.

Old Text:

We recorded the location of type-I and type-II signatures in the tunnelling current for 119 successful HR events (see Supplementary Fig. S4 for additional recordings). Figure 3c shows the distribution of distances the tip traveled towards the surface for HR to occur. The majority of events (~90%) occur before moving 550 pm. Closer tip approaches have an increased tendency to change the tip structure. This value provides an ideal parameter for automated HR, optimizing the probability of repassivation with that of harmful apex changes.

New Text:

Unlike the type-I process, which theoretically relies on a particular tip state to enable the transfer of hydrogen³⁴, the type-II process appears to be much less restrictive. We have been able to observe type-II HR events even after impressing the tip ~1 nm into the sample surface. Both processes have been observed and reproduced on a number of physically different tungsten tips, during the fabrication of numerous different structures. The structures in Fig. 2 and Fig. 4 were created using different tips for example. We recorded the location of type-I and type-II signatures in the tunneling current for 119 successful HR events collected using seven different tips (see Supplementary Fig. S4 for additional recordings). Figure 3c shows the distribution of distances the tip traveled towards the surface for HR to occur. The majority of events (~90%) occur before moving 550 pm. Closer tip approaches have an increased tendency to change the tip structure. This value provides an ideal parameter for fully automated HR, optimizing the probability of repassivation while mitigating that of harmful apex changes.

During our observations we noted that when a tip is hydrogen-functionalized, as indicated by a change in STM imaging resolution, it is still possible to transfer an off-apex hydrogen to the surface (type-II signature) without altering the apex itself (leaving the tip functionalized). That is, with a hydrogen-functionalized tip, it is not guaranteed to first remove the apex atom and observe a type-I HR signature, as an off-apex hydrogen may be more mobile and easily transferred to the surface first, causing a type-II HR signature. We have also never observed sequential type-I signatures without deliberately re-functionalizing the tip in between HR attempts, suggesting the tip is unable to functionalize spontaneously with off-apex hydrogen. This observation is consistent with experimental imaging data, where spontaneous changes in image resolution with a tip suitable for HL/HR are rare.

If the authors can provide convincing evidence here, then the work has the right caliber for publication in Nature Communications. The discussion is sound and the quality of the experimental data is very good – no doubt at this point.

We thank the reviewer for their kind appraisal of our work.

Two minor comments:

(1) Literature information of Refs. 9, 26, and 30 needs to be checked.

These errors have been corrected.

(2) In the legend of Figure S2 it is stated that “an error has been made” in a hydrogen removal step. Is it known what the resulting defect or unwanted bonding structure is in this case?

In this case, it appeared as if an error had been made because the resulting image did not show the expected dangling bond structure. Instead, a much brighter structure was in its place, often indicative that more than one hydrogen atom had been removed. However, instead of an actual error, we discovered that the removed hydrogen atom became physisorbed on the surface in close proximity to the newly created dangling bond. Moving this H atom, it was revealed that no error had actually been made. We have changed the caption to better describe this situation.

Old Text:

a, ($V=1.4$ V, $I=50$ pA, $T=4.5$ K, 1.7×8 nm²) A two DB structure with the location of the next site for HL denoted with a blue circle. **b**, The DB structure after an attempted hydrogen removal, it appears an error has been made as the site in **a**, does not resemble a single DB. **c**, After repeated scans over the structure the error was corrected and a faint object (red arrow) was now visible. The object is atomic hydrogen that physisorbed to the sample surface after extraction instead of traveling into vacuum¹. The hydrogen atom is able to sit in close proximity to the DB in **b**, without rebonding. A description of the feature in between the two DBs on the right hand side in **c**, has been proposed elsewhere³.

New Text:

a, ($V=1.4$ V, $I=50$ pA, $T=4.5$ K, 1.7×8 nm²) A two DB structure with the location of the next site for HL denoted with a blue circle. **b**, The DB structure after an attempted hydrogen removal, it appears an error has been made as the site in **a**, does not resemble a single DB, instead appearing brighter, as if multiple hydrogens have been removed from the surface. **c**, After repeated scans over the structure the error was no longer present and a faint object (red arrow) was now visible. The object is atomic hydrogen that physisorbed to the sample surface after extraction instead of traveling into vacuum¹. The hydrogen atom was altering the appearance of the newly created DB in **b**, causing it to look like an error had occurred. The hydrogen atom is able to sit in close proximity to the DB in **b**, without rebonding. A description of the feature in between the two DBs on the right-hand side in **c**, has been proposed elsewhere³.

Reviewer #3 (Remarks to the Author):

I have no hesitation in recommending this paper for publication in Nature Communications subject to very minor changes. It is a tour de force demonstration of atomic manipulation, representing a major breakthrough in an intensively studied system: STM-induced hydrogen deactivation. The authors have achieved a step change in our ability to modify matter at the atomic level: the protocols and results they describe show that single chemical bonds can now be controlled in a binary read/write fashion, surmounting a key barrier with the use of H:Si(100) as a substrate for atomic memories. More than this, however, the ability to error correct on a (dangling) bond-by-bond basis opens up exciting possibilities in the generation of cellular automata of the type previously demonstrated by the Wolow group and, more broadly, a variety of information processing (rather than solely memory) devices.

We are grateful for the positive recommendation of our work.

I recommend publication subject to the following minor changes:

(i) The reference to Michelle Simmons et al.'s work is rather old (2003). I would suggest adding some more recent papers. Please note that I am not a member of Simmons' research team, nor am I an author on any of that group's papers.

More recent papers have also been included alongside the initial reference. Reference 10 and 11 in the manuscript.

(ii) With regard to characterising the state of the probe, did the authors attempt dI/dV vs V spectroscopy to gain insights into changes in density of states? I appreciate just why this could be very difficult, in that the spectroscopy itself could modify the tip apex, but it would be helpful if the authors could add a line or two to the paper to clarify whether they think dI/dV spectroscopy (or other forms of scanning probe 'spectroscopy' such as force-distance curves) might be a viable strategy for tip characterisation.

We have not intentionally explored such an experiment to date. However, it is a great idea now that the state of the apex can be determined to no longer be hydrogen functionalized after a type-I signature is observed. Further, should dI/dV spectroscopy yield insight into the current tip state, it can serve as a more rapid metric for automated tip formation. The following discussion has been added to the text.

Old Text:

NA

New Text:

With the ability to know when a tip is hydrogen-functionalized, and when the apex atom has been removed, it may now be possible to correlate dI/dV spectroscopy curves over hydrogen-terminated silicon with the specific tip states necessary for hydrogen transfer to occur³⁴ (type-I). If such a correlation

is found, dI/dV spectroscopy would provide a new, more rapidly acquirable metric to determine if the tip is suitable for HR. These curves could then be used as training data in the automated tip forming routine to always ensure the tip is in the required state, without the need to take an entire STM image³³.

(iii) I assume that when the authors say they use a 2D Fourier transform, this was just the power spectrum? Or did they analyse the phase too? The latter might be useful in terms of characterising the probe state...

Indeed, we only deal with the power spectrum. We have clarified this within the text. The phase may also contain more detailed information on the probe state, which we do not make use of here. In the current protocol, the 2DFT is only performed at the beginning of a patterning run, as the tip state may change during the patterning process, the information gained from the phase may not be relevant through the whole process. That said, if the state of the probe could easily be extracted from the phase information of a 2DFT, it could serve in the same way as the dI/dV information described above. Such a possibility merits further investigation.

Old Text:

The HL program was designed in house, including a graphical user interface (GUI) for atomic pattern input. An artifact-free positive sample voltage STM image of the working area is first analysed to determine the position of each atom through a 2D-Fourier Transformation (2DFT), extracting the dominant spatial frequencies of the surface.

New Text:

The HL program was designed in house, including a graphical user interface (GUI) for atomic pattern input. An artifact-free positive sample voltage STM image of the working area is first analyzed to determine the position of each atom through a 2D-Fourier Transformation (2DFT), extracting the dominant spatial frequencies of the surface from the power spectrum.

Old Text:

b, 2D Fourier transform of **a**, where the three dominant spatial frequencies have been isolated.

New Text:

b, 2D Fourier transform (power spectrum) of **a**, where the three dominant spatial frequencies have been isolated.

(iv) There are a number of typos, which I assume will be picked up in the proof-reading stage.

I thoroughly enjoyed reading this paper.

Thank you!

Reviewer #4 (Remarks to the Author):

In 'Lithography for Robust, Editable Atomic-scale Silicon Devices and Memories', Roshan Achal and coworkers report on a new STM-based method to repassivate individual dangling bonds on an otherwise hydrogen passivated silicon surface. Hydrogen repassivation (HR) is a highly desired tool to complement the already much more developed hydrogen lithography (HL). Together, the two tools allow for both writing and erasing individual atomic bits. The authors demonstrate this ability by presenting, among others, a 192 bits digital music sample.

The manuscript is well written and the results are convincing and nicely presented. However, I do have some concerns as well as some suggestions for improvement of the paper. I will itemize my issues below.

We thank the reviewer for their constructive comments and suggestions.

1. Recently, HR of individual dangling bonds was already demonstrated by means of AFM (Refs. 5 and 12). The authors say about this that "the utility of this technique is limited as AFMs are not ubiquitous and are often more complex to maintain and operate". First, I don't believe that AFMs are any less ubiquitous nor harder to operate than STMs. But even if they were: if it was found that AFMs are the key to solving all the world's data storage problems, they would of course become ubiquitous soon enough. So this can hardly serve as a valid justification for having to switch to STM.

The reviewer makes a very good point regarding the current statement. We have removed it to instead focus more clearly on the differences in speed between the two systems, having used both methods. To maintain the frequency shift signal required in AFM, two feedback circuits are needed, reducing the speed at which any signal can be acquired. In STM the use of only one feedback loop simplifies operation, allowing for faster scans. Further, in the implementation here, the feedback loop is disabled altogether during HR, and the current is monitored directly, further increasing speed. As NC-AFM is able to operate in STM mode, this procedure can also complement the existing techniques. The text has been modified to better reflect these differences.

Old Text:

A means to correct errors was recently shown using a cryogenic atomic force microscope (AFM), where individual DBs were repassivated with a hydrogen functionalized tip^{5,12}. While a striking demonstration of atomic control, the utility of this technique is limited as AFMs are not ubiquitous and are often more complex to maintain and operate. Additionally, the repassivation procedure is slow (\$\sim 10\$ s/DB)¹², reducing its practicality for larger structures, further slowed by the need to re-functionalize the tip with hydrogen in between each event^{5,12}.

New Text:

A means to controllably correct errors was recently shown using a cryogenic atomic force microscope (AFM), where individual DBs were repassivated with a hydrogen-functionalized tip^{5,14}. While a striking

demonstration of atomic control, the utility of this technique is limited as the reported repassivation procedure is slow (~ 10 s per DB)¹⁴, reducing its practicality for larger structures. The frequency shift signal utilized in AFMs requires two independent feedback loops, restricting the maximum speed of the overall process. This procedure is further slowed by the need to re-functionalize the tip with hydrogen in between each event^{5,14}.

2. The other method that the current technique is compared to is the storage of data on a chlorine terminated copper surface (Ref. 2). As key difference, the authors state that with their method "the entire surface can be used for storage as the number of available bits is not predetermined at the time of sample preparation". While this may be true, they do not mention in their comparison that with the HL/HR method new hydrogen atoms need to be brought in from far away after every few bit flips. In the chlorine vacancy method this is not needed, as the bits there are encoded in the lateral position of a vacancy rather than the presence/absence of a vacancy.

We see the possible confusion with our statement. With this statement we did not intend to directly compare the read/write feasibility between the two systems. Indeed, the chlorine copper system does not require external atoms to be brought in once the bits have been written to change their state, which can be advantageous for rewriting speeds. The comparison we intended to make was the maximum theoretical surface utilization possible if the whole surface was to be used for data storage. In principle, with a sufficient supply of hydrogen, the whole Si(100):H surface could be used, whereas this is not the case in the Cl system. We have adjusted the text to clarify this point, as well as the strengths of both systems.

Old Text:

Interest in atomic memory has been reignited with foundational work on chlorine-passivated Cu(100), establishing a sophisticated scanned probe architecture to create a kilobyte of memory from surface vacancies². Despite this substantial progress, the memory is restricted to operating temperatures below 77 K. DB-based memories eliminate this restriction, as patterned structures exhibit about an order of magnitude better thermal stability^{10,11}. There are several other key advantages to DB memory blocks including a 32% larger maximum storage density. In principle the entire surface can be used for storage as the number of available bits is not predetermined at the time of sample preparation. DBs can now be created or removed as needed using HL and HR whereas creating additional vacancies/bits in the Cu(100) system is currently not possible without damaging the STM tip².

New Text:

Interest in atomic memory has been reignited with foundational work on chlorine-passivated Cu(100), establishing a sophisticated scanned probe architecture to create a kilobyte of memory from surface vacancies, without the need to vertically manipulate atoms². The memory operates near 77 K, where it remains stable for at least 44 hours². There are several key features of DB-based memories that allow us to push atomic-scale storage even further than this already substantial progress. Patterned DB structures exhibit improved thermal stability, remaining stable for an additional 400 K above liquid nitrogen temperature^{12,13}. The maximum storage density of memory blocks can be increased by 32%, as

DBs can be placed in close proximity to one another. In addition to density, the number of available bits is not predetermined at the time of sample preparation². DBs can now be created or removed as needed using HL and HR (assuming a sufficient supply of hydrogen atoms), theoretically allowing the entire hydrogen-terminated surface to be written to. Creating additional vacancies/bits in the Cu(100) system is currently not possible without damaging the STM tip².

3. Two qualitatively different HR events were recorded (Fig. 3): type I where the tip apex shape alters during HR, and type II where it remains unchanged. The authors suggest that this results from hydrogen atoms being adsorbed respectively to the apex and to the tip side. For type I the current on the downstroke is smaller than on the upstroke, while for type II this is opposite. First, it is not clear to me why the current behaves qualitatively as it does for the two types. Perhaps the authors could provide some clarification here, or at least some speculation.

More detail regarding the behavior of the current in each case has been added to the text to better illustrate why the signatures appear as they do in each case.

Old Text:

We usually observe a sudden increase in the tunneling current while approaching the surface when a DB is repassivated, that is not present as the tip retracts (Fig. 3a). The measured current is related to the overlap of the imaging orbital of the tip, and orbitals of features on the surface^{14,31}. We associate this signature (type-I) with the removal of a hydrogen at the tip apex and a return to pre-hydrogen functionalized imaging resolution.

New Text:

We usually observe a sudden increase in the tunneling current while approaching the surface when a DB is successfully repassivated that is not present as the tip retracts (Fig. 3a). The measured current is related to the overlap of the imaging orbital of the tip, and orbitals of features on the surface^{16,35}. We associate this signature (type-I) with the removal of a hydrogen at the tip apex and a return to pre-hydrogen-functionalized imaging resolution. The increase in current is possibly due to the new apex orbital having a larger spatial extent, creating a greater overlap between the tip and sample surface compared to that between a DB and a hydrogen-functionalized tip (held at the same tip-sample separation).

Old Text:

During HR, when the DB is repassivated with an off-apex hydrogen we see a second signature (type-II), a sudden decrease in tunneling current (Fig. 3b) (also observed during HR at 77 K, Supplementary Fig. S3). The sudden drop in current is due to a decrease in size of the surface orbital after the DB has been repassivated, reducing overlap/current, since the tip orbital remains unaltered.

New Text:

During HR, when the DB is re-passivated with an off-apex hydrogen we see a second signature in the tunneling current (type-II), a sudden decrease (Fig. 3b) (also observed during HR at 77 K, Supplementary Fig. S3). The sudden drop in current is due to a reduction in overlap between the tip and sample. DBs appear as bright protrusions on the sample surface (Fig. S1b) compared to the surrounding hydrogen (orbital more spatially extended towards tip). There is a decrease in the size of the surface orbital after the DB has been re-passivated, which reduces overlap/current, as the tip orbital remains unaltered.

Second, the authors report that they have acquired quite some statistics on the two HR types. My question is: after 'loading' the tip with H-atoms, is the first HR event always type I and all the subsequent events type II? Such information might help in understanding how the tip loading mechanism works and how it might be improved.

This is a very good question, we thank the reviewer for highlighting it. In our experience it is possible (though not common) to have a type-II signature precede a type-I signature when the tip is hydrogen functionalized, leaving the apex unaltered. In the case of multiple successive HR events without reloading the tip, we have not yet observed sequential type-I re-passivation signatures, indicating the tip does not readily "re-functionalize" with a hydrogen atom from the off-apex region once it has been removed. The text has been modified to include these additional details.

Old Text:

NA

New Text:

During our observations we noted that when a tip is hydrogen-functionalized, as indicated by a change in STM imaging resolution, it is still possible to transfer an off-apex hydrogen to the surface (type-II signature) without altering the apex itself (leaving the tip functionalized). That is, with a hydrogen-functionalized tip, it is not guaranteed to first remove the apex atom and observe a type-I HR signature, as an off-apex hydrogen may be more mobile and easily transferred to the surface first, causing a type-II HR signature. We have also never observed sequential type-I signatures without deliberately re-functionalizing the tip in between HR attempts, suggesting the tip is unable to functionalize spontaneously with off-apex hydrogen. This observation is consistent with experimental imaging data, where spontaneous changes in image resolution with a tip suitable for HL/HR are rare.

4. Related to the above point, I wonder to what extent the findings are truly reproducible. Often in STM, certain manipulation effects may seem very reproducible with one specific microscopic tip shape, but after the tip is deeply indented into the surface or macroscopically altered or replaced altogether, the effects do not return. Were the authors able to build useful memories with significantly different tip shapes? This question is essential in my opinion in view of possible scalability of the technique.

The maple leaf (Fig 2) and the memories (Fig 4) were created with different tips, as was figure S1. The data presented in figure 3 was collected with 7 different tips by three different STM operators. The tips themselves were also not prepared by the same person each time. We have found that it is possible to repassivate dangling bonds after indenting the tip ~ 1 nm into the surface (moving the tip 2 nm towards the surface). All of these factors lead us to believe the results are reproducible, and not reliant on a special tip, demonstrating a key aspect of scalability. The text has been updated to include some of these details.

Old Text:

We recorded the location of type-I and type-II signatures in the tunnelling current for 119 successful HR events (see Supplementary Fig. S4 for additional recordings). Figure 3c shows the distribution of distances the tip traveled towards the surface for HR to occur. The majority of events ($\sim 90\%$) occur before moving 550 pm. Closer tip approaches have an increased tendency to change the tip structure. This value provides an ideal parameter for automated HR, optimizing the probability of repassivation with that of harmful apex changes.

New Text:

Unlike the type-I process, which theoretically relies on a particular tip state to enable the transfer of hydrogen³⁴, the type-II process appears to be much less restrictive. We have been able to observe type-II HR events even after impressing the tip ~ 1 nm into the sample surface. Both processes have been observed and reproduced on a number of physically different tungsten tips, during the fabrication of numerous different structures. The structures in Fig. 2 and Fig. 4 were created using different tips for example. We recorded the location of type-I and type-II signatures in the tunneling current for 119 successful HR events collected using seven different tips (see Supplementary Fig. S4 for additional recordings). Figure 3c shows the distribution of distances the tip traveled towards the surface for HR to occur. The majority of events ($\sim 90\%$) occur before moving 550 pm. Closer tip approaches have an increased tendency to change the tip structure. This value provides an ideal parameter for fully automated HR, optimizing the probability of repassivation while mitigating that of harmful apex changes.

5. The importance of the Fourier analysis method is not quite clear to me. I can understand that such Fourier filtering of the images would help for automated pattern recognition, but is it really a key element in the current finding? The authors devote an entire figure in the main manuscript to it (Fig. 1), which could also be used for some of the analysis suggested above.

We hope to present a lithographic process in this work that can be easily implemented by interested groups. Accurate autonomous patterning without properly characterizing the surface for alignment is almost impossible. Slight errors in tip position can result in the incorrect atoms being removed. The FT method allows us to determine the position of each atom and travel site to site without the need to reimage the surface each time, yielding more accurate and faster automation. Further, each STM motor is different (in the x and y direction for example) and may move a different amount to identical applied voltages. These non-linearities are directly accounted for in the FT, reducing this possible source of error

dramatically. Without such a process, intricate patterns such as in figure 2 could not be readily achieved, and the techniques here would be difficult to scale. We have added additional text to emphasize the importance of the FT method.

Old Text:

To begin automated HL, the location of every hydrogen atom in a select area is determined for accurate STM tip registration during fabrication. The periodicity of the hydrogen passivated Si(100)-2x1 surface (Fig. 1a) permits the location of every hydrogen atom to be determined from a single STM image (while accounting for nonlinearities in the scanner) through the use of Fourier analysis²⁸ (Fig. 1b-f). Once the surface has been characterized, the desired pattern is mapped onto the lattice. Next, the tip is brought over each lattice site of the pattern and 20 ms voltage pulses between 1.8 to 3.0 V are applied at a fixed tip-height ($V=1.4$ V, $I=50$ pA) until the successful removal of hydrogen has been detected. Unlike conventional HL^{14,29}, the STM feedback control is disabled during the voltage pulses. This allows jumps in the tunneling current to be used as an indicator of success¹³, which can be detected faster and more accurately than jumps in tip-height through the feedback circuitry (see Methods). Figures 2a,b,d show the process of building DB structures (Fig 2e) with HL, while using HR to correct errors (Fig 2c).

New Text:

To begin automated HL, the location of every hydrogen atom in a select area needs to be determined for accurate STM tip registration during fabrication. Slight errors in the tip position can result in incorrect atoms being removed. Fast, fully autonomous lithography also requires the location of each atom to be known, such that the surface doesn't need to be reimaged after each removal event to determine the next site. The periodicity of the hydrogen-passivated Si(100)-2x1 surface (Fig. 1a) permits the location of every hydrogen atom to be determined from a single STM image (while accounting for nonlinearities in the scanner) through the use of Fourier analysis³¹ (Fig. 1b-f). Such an analysis is relatively immune to the presence of small surface defects and dangling bonds due to their spatially localized nature in the images compared to the extended periodicity of the surface itself. Figure 1 illustrates the basic features of this process. In practice, we use images between 10×10 nm² to 40×40 nm² to determine the location of the hydrogen atoms on a given sample terrace. Once the surface has been characterized, the desired pattern is mapped onto the lattice. Next, the tip is brought over each lattice site of the pattern and 20 ms voltage pulses between 1.8 and 3.0 V are applied at a fixed tip-height ($V=1.4$ V, $I=50$ pA) until the successful removal of hydrogen has been detected. Figures 2a,b,d show the process of building DB structures (Fig 2e) with HL while using HR to correct errors (Fig 2c). Unlike conventional HL^{16,32}, the STM feedback control is disabled during the voltage pulses. This allows jumps in the tunneling current to be used as an indicator of success¹⁵, which can be detected faster and more accurately than jumps in tip-height through the feedback circuitry (see Methods).

In order to make the manuscript suited for publication in Nature Communication, I believe that at least the issues above should be addressed.

REVIEWERS' COMMENTS:

Reviewer #1 (Remarks to the Author):

I thank the Authors for detailed answers for all my comments. The revised manuscript has been much improved in terms of the technical details description and in my opinion it is now suitable for publication in Nature Comm.. However, I am bit surprised that the extended discussion in the revised manuscript is not directly supported by any additional experimental data, which could clarify my concerns about scalability of the proposed protocols. Thus I still keep my personal feeling that this work is not an experimental or conceptual break through. That could eventually limit its future impact.

Reviewer #2 (Remarks to the Author):

2nd Report – NCOMMS-18-05289

The authors have addressed the points raised in my previous report. After revision, the manuscript now contains additional information that supports the reliability and robustness of the described HL fabrication technique in a convincing way. The authors also provided a detailed point by point response to the reports of the other reviewers. I therefore recommend publication of the manuscript in Nature Communications in its present form.

Reviewer #4 (Remarks to the Author):

The authors have done a very thorough job in addressing all the issues raised by me and the other reviewers. Regarding my own concerns, I am now more convinced of the authors' findings than before. Specifically the fact that they report similar behaviour with multiple microscopically different tips is to me a key requirement that has now been covered.

With regard to the remarks of the remaining reviewers, I leave it to their judgement if their issues have been properly dealt with. So I would support the manuscript for publication in Nature Communication, pending the other reviewers' approval.

Reviewer #1 (Remarks to the Author):

I thank the Authors for detailed answers for all my comments. The revised manuscript has been much improved in terms of the technical details description and in my opinion it is now suitable for publication in Nature Comm.. However, I am bit surprised that the extended discussion in the revised manuscript is not directly supported by any additional experimental data, which could clarify my concerns about scalability of the proposed protocols. Thus I still keep my personal feeling that this work is not an experimental or conceptual break through. That could eventually limit its future impact.

Reviewer #2 (Remarks to the Author):

The authors have addressed the points raised in my previous report. After revision, the manuscript now contains additional information that supports the reliability and robustness of the described HL fabrication technique in a convincing way. The authors also provided a detailed point by point response to the reports of the other reviewers. I therefore recommend publication of the manuscript in Nature Communications in its present form.

Reviewer #4 (Remarks to the Author):

The authors have done a very thorough job in addressing all the issues raised by me and the other reviewers. Regarding my own concerns, I am now more convinced of the authors' findings than before. Specifically the fact that they report similar behaviour with multiple microscopically different tips is to me a key requirement that has now been covered.

We thank the reviewers for their recommendation for publication, and their comments and questions. The manuscript is now more clear and accessible as a result.